# IASI nitrous oxide $(N_2O)$ retrievals: validation and application to transport studies at daily time scales

Yannick Kangah<sup>1</sup>, Philippe Ricaud<sup>1</sup>, Jean-Luc Attié<sup>2</sup>, Naoko Saitoh<sup>3</sup>, Jérôme Vidot<sup>4</sup>, Pascal Brunel<sup>4</sup>, and Samuel Quesada-Ruiz<sup>1</sup>

<sup>1</sup>CNRM-GAME, Météo-France/CNRS UMR 3589, Toulouse, France

<sup>2</sup>Université de Toulouse, Laboratoire d'Aérologie, CNRS UMR 5560, Toulouse, France

<sup>3</sup>Center for Environmental Remote Sensing, Chiba University, Japan

<sup>4</sup>Centre de Météorologie Spatiale,DP/Météo-France, Lannion, France

Correspondence to: Yannick Kangah (yannick.kangah@umr-cnrm.fr)

Abstract. The aim of this paper is to present a method to retrieve nitrous oxide (N<sub>2</sub>O) vertical profiles from the Infrared Atmospheric Sounding Interferometer (IASI) onboard the MetOp platform. We retrieved N<sub>2</sub>O profiles using IASI clear sky radiances in 2 spectral bands: B1 and B2 centered at  $\sim$ 1280 cm<sup>-1</sup> and  $\sim$ 2220 cm<sup>-1</sup>, respectively. Both retrievals in B1 and B2 (hereafter referred to as N<sub>2</sub>O\_B1 and N<sub>2</sub>O\_B2, respectively) are sensitive to the mid-to-upper troposphere with a

- maximum of sensitivity at around 309 hPa. The degrees of freedom for N<sub>2</sub>O\_B1 and N<sub>2</sub>O\_B2 are 1.38 and 0.93, respectively. We validated the retrievals using the High-performance Instrumented Airborne Platform for Environmental Research Pole-to-Pole Observations (HIPPO). The comparisons between HIPPO and the two retrieved datasets show relatively low standard deviation errors around 1.5% (~4.8 ppbv) and 1.0% (~3.2 ppbv) for N<sub>2</sub>O\_B1 and N<sub>2</sub>O\_B2, respectively. However, the impact of H<sub>2</sub>O contamination on N<sub>2</sub>O B1 due to its strong absorption bands in B1 significantly degrades the quality of the retrievals in
- tropical regions. We analysed the scientific consistency of the retrievals at 309 hPa with a focus on the long-range transport of N<sub>2</sub>O especially during the Asian summer monsoon. Over the mid-latitude regions, both variations of N<sub>2</sub>O\_B1 and N<sub>2</sub>O\_B2 at 309 hPa are influenced by the stratospheric N<sub>2</sub>O-depleted air because of the relative coarse shape of the averaging kernel. The analysis of N<sub>2</sub>O\_B2 using results from backtrajectories exhibits the capacity of these retrievals to capture long-range transport of air masses from Asia to northern Africa via the summer monsoon anticyclone on a daily basis. Thus, N<sub>2</sub>O\_B1 and N<sub>2</sub>O\_B2
- offer an unprecedented possibility to study global upper tropospheric  $N_2O$  on a daily basis.

# 1 Introduction

Nitrous oxide (N<sub>2</sub>O) is a long-lived greenhouse gas with a lifetime of about 120 years which is essentially produced in the terrestrial and oceanic surfaces by the microbial processes of nitrification and denitrification (Butterbach-Bahl et al., 2013). In terms of radiative forcing, N<sub>2</sub>O is the third anthropogenic greenhouse gas after methane (CH<sub>4</sub>) and carbon dioxide (CO<sub>2</sub>)

(Ciais et al., 2014). Its main sink is the photolysis in the stratosphere but it is also destroyed by reacting with the excited atomic oxygen  $O(^{1}D)$ . This reaction is the main source of the nitrogen oxides, which are the main responsible of the destruction of the stratospheric ozone. N<sub>2</sub>O is therefore becoming the main ozone depleting substance emitted in the 21<sup>st</sup> century (Ravishankara

et al., 2009). The natural and anthropogenic  $N_2O$  emissions are about 60% and 40%, respectively (Syakila and Kroeze, 2011; Bouwman et al., 2013). The anthropogenic  $N_2O$  emissions are dominated by agricultural sources which represent more than 66% of these emissions. An increase of the  $N_2O$  volume mixing ratio (vmr) with a mean rate of 0.75 ppbv.yr<sup>-1</sup> since the late 1970s has been observed (Ciais et al., 2014). This positive trend is driven by anthropogenic emissions because of the increasing use of nitrogen fertilizers to meet the growing demand of food production, especially in Asia. Moreover, according to the

Intergovernment Panel on Climate Change (IPCC), this trend is likely to continue until 2100. Monitoring N<sub>2</sub>O emissions and its atmospheric concentration are therefore becoming major issues in the framework of anthropogenic pollution mitigation. Nowadays, surface measurements of N<sub>2</sub>O provide the longer time series of N<sub>2</sub>O measurements and are used to character-

ize the trends and the sources of tropospheric  $N_2O$ . Such measurements are performed by several organizations or institutes

- such as the National Oceanic and Atmospheric Administration/Earth Systems Research Laboratory/Global Monitoring Division (NOAA/ESRL/GMD) or in the framework of joint projects such as the Advanced Global Atmospheric Gases Experiment (AGAGE) (Ganesan et al., 2015) and the Network for the Detection of Atmospheric Composition Change (NDACC) (http://www.ndsc.ncep.noaa.gov). Despite their reliability and the long-term records of surface measurements, their limited geographical coverage makes them difficult to use in order to assess N<sub>2</sub>O tropospheric variations at global scale. In addition to
- surface measurements, there are also some aircraft campaigns like the High-performance Instrumented Airborne Platform for Environmental Research Pole-to-Pole Observations (HIPPO) (Wofsy, 2011; Wofsy et al., 2012) over the Pacific Ocean. N<sub>2</sub>O is also measured in some passenger aircraft based measurements including the Comprehensive Observation Network for TRace gases by AIrlLner (CONTRAIL) (Sawa et al., 2015) and the Civil Aircraft for the Regular Investigation of the atmosphere Based on an Instrument Container (CARIBIC) (Assonov et al., 2013).
- Since satellite measurements of stratospheric N<sub>2</sub>O began in the 1970s, tropospheric N<sub>2</sub>O retrievals using satellite measurements are relatively recent. Clerbaux et al. (2009) exhibit the N<sub>2</sub>O signature from the infrared measurements of the Infrared Atmospheric Sounding Interferometer (IASI) showing some promising results in view of using these measurements to retrieve N<sub>2</sub>O tropospheric profiles. Ricaud et al. (2009a) analysed the equatorial maximum of N<sub>2</sub>O during March-May using the operational total columns of N<sub>2</sub>O retrieved from IASI measurements using artificial neural networks. These operational N<sub>2</sub>O
- total column products also show seasonal cycles and annual trends consistent with the retrieved N<sub>2</sub>O from the ground-based Fourier Transform Spectrometer (FTS) observations at the Izaña Atmospheric Observatory (IZO, Spain) (García et al., 2014, 2016). First results of N<sub>2</sub>O total columns retrievals using a partially scanned IASI interferogram with an accuracy of  $\pm 13$  ppbv (~4%) are described in Grieco et al. (2013). Retrievals of N<sub>2</sub>O tropospheric profiles have been performed using the Atmospheric Infrared Sounder (AIRS) and the results showed interannual trends consistent with surface measurements (Xiong et al.,
- 2014). N<sub>2</sub>O profiles retrieved from the Greenhouse Gas Observing Satellite (GOSAT) measurements have been used to study the transport of Asian summertime high N<sub>2</sub>O emissions to the Mediterranean upper troposphere (Kangah et al., 2017).

In this paper, we describe the IASI instrument and the Radiative Transfer for Tiros Operational Vertical sounder (RTTOV) used as forward model in our retrieval system in sections 2 and 3, respectively. We present the retrieval strategy and the validation of the results using HIPPO airborne in situ measurements in sections 5 and 6, respectively. In section 7, we analyse

the scientific consistency of the retrievals focusing on the long-range transport of N<sub>2</sub>O during the Asian summer Monsoon

using backtrajectories from the Hybrid Single-Particle Lagrangian Integrated Trajectory model (HYSPLIT) model (Stein et al., 2015). Conclusions are presented in section 8.

## 2 IASI

- IASI is a spaceborne instrument on board the platforms MetOp-A and MetOp-B. The MetOp (Meteorological Operational) mission consists of a series of three sun-synchronous Low Earth Orbits satellites developed jointly by the french space agency (CNES) and the EUropean organization for the exploitation of METeorological SATellites (EUMETSAT). The first satellite (MetOp-A) was launched in October 2006, the second (MetOp-B) in September 2012 and the third (MetOp-C) is expected to be launched in October 2018. MetOp-A and MetOp-B are operational at the present time. The mean MetOp altitude is ~820 km and the satellite crosses the equator at ~09:30 mean local solar time and have a repeat cycle of 29 days. MetOp-A and
- MetOp-B are in the same orbital plane and have an orbit phasing of about 49 min. IASI is a Michelson interferometer that measures infrared spectrum in the spectral range from 645 to 2760 cm<sup>-1</sup> (15.5 to 3.62  $\mu$ m) (Clerbaux et al., 2009). Although its apodized spectral resolution is about 0.5 cm<sup>-1</sup>, IASI provides each spectrum with a sampling of 0.25 cm<sup>-1</sup> giving a total of 8461 channels. The large spectral domain of IASI contains absorption bands of several atmospheric constituents (Hilton et al., 2012) among which the major absorbers are water vapour (H<sub>2</sub>O), ozone (O<sub>3</sub>), CO<sub>2</sub>, N<sub>2</sub>O, CH<sub>4</sub> and carbone monoxide
- (CO). IASI observes the Earth with a swath of about 2200 km (1100 km on each side) and its instantaneous field of view is composed of four circular pixels of 12 km diameter footprint on the ground at nadir. The operational IASI H<sub>2</sub>O, temperature and O<sub>3</sub> products are retrieved simultaneously using an optimal estimation method (Pougatchev et al., 2009; Rodgers et al., 2000) whereas total columns of the other molecules are retrieved using artificial neural networks (Turquety et al., 2004). In this work, we used the IASI level 1c spectra (calibrated and apodized spectra) to perform our retrievals.

## 20 **3 RTTOV**

RTTOV is a fast model of transmittances of the atmospheric gases that are generated from a database of accurate line-byline (LBL) transmittances (Saunders et al., 1999). The database of accurate transmittances is generated from a set of diverse atmospheric profiles and then a linear regression is computed linking the optical depths of the vertical layers and a set of atmospheric profile-dependent predictors. The regression coefficients are actually given for different Instrument Spectral Response

- Functions (ISRF) including the ones of IASI. For our retrieval system, we used RTTOV version 11.2 together with the regression coefficients v9 based on the model LBLRTM (LBL Radiative Transfer Model) (Hocking et al., 2015). In this version, the predictors depend on the trace gases profiles including H<sub>2</sub>O, O<sub>3</sub>, CO<sub>2</sub>, N<sub>2</sub>O, CH<sub>4</sub> and CO. It takes less than 25 ms to compute 183 IASI channels together with weighting functions using an input of atmospheric profiles on 54 vertical levels and surface emissivities. Comparing with accurate LBL models, the biases of RTTOV simulations for IASI Brightness Temperature (BT)
- over sea in clear sky conditions are within  $\pm 1$  K in the spectral range 645 to 2000 cm<sup>-1</sup> and within  $\pm 1.6$  K in the N<sub>2</sub>O/CO<sub>2</sub>  $\nu$ 3 region between 2200 and 2300 cm<sup>-1</sup> (Matricardi, 2009).

## 4 N<sub>2</sub>O absorption bands

Previous studies from Clerbaux et al. (2009) have highlighted three absorption bands of N<sub>2</sub>O in the IASI spectral range centered at ~1280 cm<sup>-1</sup>, ~2220 cm<sup>-1</sup> and ~2550 cm<sup>-1</sup>. Figure 1 shows a N<sub>2</sub>O weighting function matrix (called hereafter Jacobian matrix) calculated in units of brightness temperature (BT) using a N<sub>2</sub>O profile derived from the Michelson Interferometer for
Passive Atmospheric Sounding (MIPAS) reference atmosphere (V3) daytime mid-latitude climatology. This matrix represents the sensitivity of the calculated BT to a unit change in the N<sub>2</sub>O volume mixing ratio (vmr). The spectral signature of N<sub>2</sub>O appears in the three spectral regions with significant differences of intensity. The most intense absorption band (called hereafter B2) is between 2190 and 2240 cm<sup>-1</sup> and shows sensitivity to N<sub>2</sub>O from the lowermost troposphere to 100 hPa with a maximum of sensitivity between 500 and 200 hPa. The absorption band located between 1250 and 1310 cm<sup>-1</sup> (called hereafter B1) is less
intense than B2 and is sensitive to N<sub>2</sub>O between 800 and 100 hPa. The third band (called hereafter B3) located between 2500 and 2600 cm<sup>-1</sup> is much less intense than B1 and B2 and is sensitive to N<sub>2</sub>O from 900 to 300 hPa. To illustrate the sensitivity of

- and  $2600 \text{ cm}^{-1}$  is much less intense than B1 and B2 and is sensitive to N<sub>2</sub>O from 900 to 300 hPa. To illustrate the sensitivity of these three bands to N<sub>2</sub>O and to the other atmospheric and surface parameters, a sensitivity study has been performed using the MIPAS climatology for N<sub>2</sub>O, CO<sub>2</sub> and O<sub>3</sub> profiles and a set of atmospheric and surface parameters representative of a given atmospheric state on 13 June 2011 at 11.8°N and 142.9°W. This study consists in calculating of the variation of the BT (called
- hereafter  $\Delta BT$ ) over the IASI spectral range for a given variation of the major atmospheric and surface parameters consistent with their actual accuracy. Figure 2 shows the absolute value of the  $\Delta BT$  ( $|\Delta BT|$ ) for variations of each major absorber (H<sub>2</sub>O, O<sub>3</sub>, CO<sub>2</sub>, N<sub>2</sub>O, CH<sub>4</sub> and CO) and for variations of temperature and surface temperature. The IASI radiometric noise expressed as the Noise Equivalent Delta Temperature (NEDT) is superimposed to the  $|\Delta BT|$  signals. In each band, channels were selected by optimizing the Signal to Noise Ratio (SNR) while reducing the spectral signature of the other parameters. B1
- is mainly impacted by temperature, H<sub>2</sub>O, CH<sub>4</sub> and surface temperature. The signal corresponding to 10% change of H<sub>2</sub>O is more than twice greater than the signal corresponding to a change of N<sub>2</sub>O by 4% in most spectral domains of B1. The signal corresponding to a change of CH<sub>4</sub> by 2% is half the size to the signal of N<sub>2</sub>O. A total of 126 channels is selected in B1. The signal of N<sub>2</sub>O is twice larger than the NEDT for all selected channels in B1 whilst CH<sub>4</sub>, H<sub>2</sub>O and temperature are critical parameters for the N<sub>2</sub>O retrieval using the 126 selected channels in B1. In B2, we selected a total of 103 channels where
- the signal of N<sub>2</sub>O is more than twice greater than the signals of the other parameters except for atmospheric temperature and NEDT. The NEDT level of magnitude is similar to the signal of N<sub>2</sub>O while the  $|\Delta BT|$  signal corresponding to the temperature variation is slightly greater than that of N<sub>2</sub>O. The radiometric noise and the atmospheric temperature are therefore the critical parameters for the N<sub>2</sub>O retrieval in B2. In B3, we selected no channels because the radiometric noise is too large compared to the signal of N<sub>2</sub>O. In summary, the absorption band of N<sub>2</sub>O in B2 is sufficiently isolated from the absorption band of the other
- gases but presents the same level of magnitude as the IASI radiometric noise whereas in B1 the signal of  $N_2O$  is more than twice greater than the noise but is impacted by the absorption bands of  $CH_4$  and  $H_2O$ .

## 5 Retrieval Strategy

#### 5.1 Methodology

We used an optimal estimation method based on the Levenberg-Marquardt iterative algorithm (Rodgers et al., 2000) to retrieve N<sub>2</sub>O profiles over 13 fixed pressure levels from IASI clear sky radiances in the bands B1 and B2. Hereafter, the retrievals in B1 and B2 are referred to N<sub>2</sub>O\_B1 and N<sub>2</sub>O\_B2, respectively. In the retrieval algorithm, the i+1<sup>th</sup> retrieval vector is expressed as:

$$\hat{X}_{i+1} = X_a + (K_i^T S_y^{-1} K_i + \gamma S_a^{-1})^{-1} \times \{K_i^T S_y^{-1} ([Y - F(\hat{X}_i)] + K_i [\hat{X}_i - X_a]) + \gamma S_a^{-1} [\hat{X}_i - X_a]\}$$
(1)

where  $X_a$  is an a priori vector with an error covariance matrix  $S_a$ . Y is the observed radiances with an error covariance matrix  $S_y$ .  $F(\hat{X}_i)$  and  $K_i$  are the calculated forward spectrum and the Jacobian matrix at the iteration *i*, respectively.  $\gamma$  is the Levenberg-Marquardt parameter (Rodgers et al., 2000). The vertical sensitivity of the retrieval can be characterised using the

averaging kernel matrix (A) defined as:

10

$$A = \frac{\partial \hat{X}}{\partial X} = (K^T S_y^{-1} K + S_a^{-1})^{-1} K^T S_y^{-1} K$$
(2)

N<sub>2</sub>O\_B1 profiles are retrieved simultaneously with the vmr profiles of H<sub>2</sub>O and CH<sub>4</sub> whilst N<sub>2</sub>O\_B2 profiles are retrieved simultaneously with the vmr profiles of H<sub>2</sub>O, CO and CO<sub>2</sub>. The air temperature profiles and the surface parameters (temperature and emissivity) are also retrieved simultaneously with the N<sub>2</sub>O profiles for N<sub>2</sub>O\_B1 and N<sub>2</sub>O\_B2.

The a priori error covariance matrix  $S_a$  is calculated as follows:

$$Sa_{ij} = \sigma_a^2 \times exp(-|ln(P_i) - ln(P_j)|) \tag{3}$$

where  $\sigma_a^2$  is an a priori variance error fixed for each parameter of the state vector and  $P_i$  the pressure level at the level *i*.

For the retrievals, we used a fixed N<sub>2</sub>O a priori profile derived from the MIPAS V3 reference atmosphere daytime mid-20 latitude climatology. Since this climatology is given for the year 2001, we adjusted it for the year 2011 by applying the averaged increase rate of 0.75 ppbv.yr<sup>-1</sup> consistently with Ricaud et al. (2009b). We fixed  $\sigma_a$  for the N<sub>2</sub>O profile to 4% consistently with Grieco et al. (2013).

The a priori states of H<sub>2</sub>O, temperature and surface temperature were taken from the IASI level 2 operational products (August et al., 2012). A validation using radiosonde data gave a standard error (std) of  $\sim$ 2 K for the surface temperature, of

about 10% for the relative humidity and between 0.6 and 1.5 K for the temperature profile (Pougatchev et al., 2009). Thus, we took for N<sub>2</sub>O\_B1 and N<sub>2</sub>O\_B2,  $\sigma_a$  values of 1 K and 2 K for the temperature profile and the surface temperature, respectively. A  $\sigma_a$  value of 10% for the H<sub>2</sub>O profile was used for N<sub>2</sub>O\_B2. Clerbaux et al. (2009) show the presence of a relatively strong absorption band of the deuterium hydrogen oxide (HDO) also called semiheavy water in the band B1. However, this chemical

species is not taken into account as a variable parameter in RTTOV. Therefore, after sensitivity studies, we fixed the  $\sigma_a$  value for the H<sub>2</sub>O profile to 30% for N<sub>2</sub>O\_B1. In a similar approach to the N<sub>2</sub>O a priori profile, we took the CO<sub>2</sub> a priori from the MIPAS reference atmosphere v3 daytime mid-latitude climatology and applied an annual trend of 2.3 ppmv.yr<sup>-1</sup> (Ciais et al., 2014). A  $\sigma_a$  of 2% is used for the CO<sub>2</sub> a priori profile after a sensitivity study.

- In addition,  $CH_4$  and CO a priori profiles were taken from the Monitoring Atmospheric Composition and Climate (MACC) project reanalysis (Inness et al., 2013).  $\sigma_a$  was fixed to 10% for CO after a sensitivity study and consistently with the CO validation reports (http://www.gmes-atmosphere.eu/services/aqac/global\_verification/validation\_reports/). For  $CH_4$ ,  $\sigma_a$  was fixed to 2% which is approximately the std error on the IASI retrieved  $CH_4$  profiles (Xiong et al., 2013). The land surface a priori emissivity is derived from a global atlas of land surface emissivity based on inputs from the Moderate Resolution Imaging
- Spectroradiometer (MODIS) operational product (Borbas and Ruston, 2010; Seemann et al., 2008). Over sea surface, we used the version 6 of the Infrared Surface Emissivity Model (ISEM) (Sherlock and Saunders, 1999) as an a priori surface emissivity.  $\sigma_a$  is fixed to 10% for the surface emissivity since this parameter is also used as a sink parameter. An observation error diagonal covariance matrix  $S_y$  was used for the retrievals in both bands with the IASI radiometric noise as the diagonal elements of the matrix.

#### 15 5.2 Data quality control

To assess the quality of the retrieved N<sub>2</sub>O profiles, we used quality parameters derived from the optimal estimation theory (Rodgers et al., 2000). Our retrieval process consists in the minimization of the cost function  $\chi^2$  defined as:

$$\chi^{2} = \frac{\left([\hat{X} - X_{a}]^{T} S_{a}^{-1} [\hat{X} - X_{a}]\right) + \left([Y - F(\hat{X})]^{T} S_{y}^{-1} [Y - F(\hat{X})]\right)}{dim(\hat{X}) + dim(Y)}$$
(4)

where  $dim(\hat{X})$  and dim(Y) are the dimensions of the state vector and of the radiances (number of channels), respectively.  $\chi^2$ 

a n

allows to evaluate the quality of the retrieval by combining the calculated residuals relative to the observations error covariance matrix and the difference between the estimated and the a priori profiles relative to the a priori error covariance matrix. In our case, we performed simultaneous retrievals for both N<sub>2</sub>O\_B1 and N<sub>2</sub>O\_B2. Therefore, the  $\chi^2$  derived from the optimal estimation theory is a quality control parameter for the whole retrieved state vectors which include N<sub>2</sub>O profiles and the other interfering parameters. In addition to  $\chi^2$ , we computed another variable to assess the quality of the retrieved tropospheric N<sub>2</sub>O

profile which is our target species. Thus, we calculate the difference between the a priori and the retrieved N<sub>2</sub>O relative to the N<sub>2</sub>O a priori errors  $\sigma_a$ . This variable called  $\chi^2_{N_2O}$  is defined as:

$$\chi^2_{N_2O} = \frac{\sum_{P_j < 1000 \ hPa}^{P_j > 200 \ hPa} [\hat{X}_j - X_{a_j}]^2 \beta_{a_j}}{n_p} \tag{5}$$

where  $\hat{X}_j$  and  $X_{a_j}$  are the retrieved and the a priori N<sub>2</sub>O at the pressure level  $P_j$ , respectively.  $\beta_{a_j}$  is the diagonal element of the a priori error precision matrix (the inverse of the a priori error covariance matrix) at the pressure level  $P_j$  and  $n_p$  is the

number of levels used for the calculation.

An upper limit for the  $\chi^2$  parameter is generally used to select good quality pixels. For instance, an upper limit of 3 on a  $\chi^2$  calculated in the radiances space was used to select good quality pixels for CH<sub>4</sub> retrievals from IASI measurements (Xiong et al., 2013). Following the same methodology, we applied an upper limit on  $\chi^2_{N_2O}$  to select good quality pixels. After performing sensitivity studies for both N<sub>2</sub>O\_B1 and N<sub>2</sub>O\_B2, we rejected all the data with a  $\chi^2$  or a  $\chi^2_{N_2O}$  greater than or equal to 4.

Moreover, to evaluate the impact of the other retrieved parameters on  $N_2O_B1$  and  $N_2O_B2$ , we calculated the Contamination Factor (called hereafter CF) defined as follows:

$$CF(i) = \sum_{j} \left| \frac{\partial \hat{x}_{i}}{\partial c_{j}} \right| \frac{\Delta c_{j}}{\hat{x}_{i}} \times 100$$
(6)

- Here, CF(i) is the contamination of the parameter c on the retrieved N<sub>2</sub>O at the level i ( $\hat{x}_i$ ).  $\Delta c_j$  is the uncertainty on the parameter c at the level j. We fixed  $\Delta c_j$  to the a priori error  $\sigma_a$  for each parameter. Then for the parameter c, we defined  $CF_{tot}(c)$  as the sum of the CF over the 13 retrieval levels. CF indicates the influence of the uncertainties in the knowledge of the co-retrieved parameters on the variability of the target species N<sub>2</sub>O retrievals. Here, the uncertainties on the co-retrieved parameters have been fixed to the a priori uncertainties. Thus, CF does not take into account the effects due to the spatial and
- temporal variations of these uncertainties. But CF estimates, a priori, how critical is the characterisation of each co-retrieved parameter for the quality of the N<sub>2</sub>O retrievals. As consequence, a posteriori sensitivity studies should be performed on each critical parameter to determine which co-retrieved parameters uncertainties have the most significant impact on the quality of the N<sub>2</sub>O retrievals.

## 6 Validation

- In this section, we analyse the performance of our retrieval system by comparing the results with the in-situ measurements from the five HIPPO airborne campaigns (Figure 3): HIPPO 1 (January 2009), HIPPO 2 (October-November 2009), HIPPO 3 (March-April 2010), HIPPO 4 (June-July 2011) and HIPPO 5 (August-September 2011). For this purpose, we processed 26850 N<sub>2</sub>O\_B1 and N<sub>2</sub>O\_B2 profiles along the flight paths from the five HIPPO campaigns. Using a similar method as explained in Kangah et al. (2017), we used for these comparisons the measurements from the Harvard/Aerodyne Quantum Cascade
- Laser Spectrometer (QCLS), one of the airborne instruments of HIPPO, and the retrieved profiles selected within a collocation temporal and spatial window of  $\pm 200$  km and  $\pm 12h$ , respectively. Our aim is to characterise the retrieval errors as well as the ability of the retrieval system to capture N<sub>2</sub>O tropospheric variations.

#### 6.1 Error Characterisation

The total retrieval errors can be divided into four components: a smoothing error, a forward model error, a model parameter 30 error and a retrieval noise. We used a simultaneous retrieval strategy to include all the parameters which influence RTTOV in

each band and we removed RTTOV systematic biases consistently with Matricardi (2009). Therefore, the forward model and the model parameter errors can be, as a first approximation, considered as negligible compared to the smoothing error and the retrieval noise. The covariance matrix of the smoothing error  $(S_s)$  is defined as:

$$S_s = (A - I)S_e(A - I)^T$$
<sup>(7)</sup>

5

where A is the N<sub>2</sub>O averaging kernels matrix; I is the identity matrix and  $S_e$  is the covariance matrix of the real ensemble of states consistently with Rodgers et al. (2000). For our retrieval algorithm, we use a simple "ad hoc" matrix (see Eq. 3) as a priori covariance matrix ( $S_a$ ) to constrain the retrieval system. Since this matrix may or may not be representative of the variability of a real ensemble of N<sub>2</sub>O profiles, we took  $S_e$  as the covariance matrix of HIPPO profiles. The retrieval noise covariance matrix ( $S_n$ ) is defined as:

$$10 \quad S_n = G S_y G^T \tag{8}$$

where G is the gain matrix which represents the change in the vmr profile for a unit change in the observation Y. The theoretical covariance matrix of the total errors  $(S_{tot})$  is therefore defined as:

$$S_{tot} = S_s + S_n \tag{9}$$

- The theoretical covariance matrix of the total errors is then compared with an empirical total errors covariance matrix 15 calculated using the HIPPO measurements and the retrievals along the HIPPO campaigns flight paths (namely the covariance matrix of the difference between HIPPO profiles and IASI retrieved profiles). Figure 4 shows the standard deviation errors (std errors) corresponding to all these covariance matrices (square roots of the diagonal elements of the covariance matrix) and averaged over the set of retrievals for N<sub>2</sub>O\_B1 and N<sub>2</sub>O\_B2. The empirical std error which we consider as our reference standard deviation of the total errors ( $\sigma_{tot}$ ) is about 1.5% (~4.8 ppbv) for N<sub>2</sub>O\_B1 and about 1.0% (~3.2 ppbv) for N<sub>2</sub>O\_B2
- 20 in the troposphere. For N<sub>2</sub>O\_B2, the theoretical  $\sigma_{tot}$  is consistent with the empirical  $\sigma_{tot}$  but, for N<sub>2</sub>O\_B1, the theoretical  $\sigma_{tot}$  is about 0.5% less than the empirical  $\sigma_{tot}$ . This means that our hypothesis of two sources of errors to characterise the total error is correct for N<sub>2</sub>O\_B2 but is not enough for N<sub>2</sub>O\_B1 for which other sources of error should be considered (forward model errors and/or model parameter errors). Concerning the forward model errors, we removed the biases on RTTOV IASI clear sky radiances consistently with Matricardi (2009) both in the band B1 and B2. Therefore the difference between the theoretical
- and the empirical std errors for N<sub>2</sub>O\_B1 is certainly due to the existence of other sources of variation of the radiances in the band B1 which are not correctly taken into account in our retrieval system. The HDO absorption which is the only significant absorption band not included in the predictor parameters of RTTOV could be responsible of at least part of these unexplained variations. To summarise, we can consider that the std errors on N<sub>2</sub>O\_B1 and N<sub>2</sub>O\_B2 are on averaged about 1.5% (~4.8 ppbv) and 1.0% (~3.2 ppbv), respectively. However, for the users, the retrieved profiles will be given with the empirical  $S_{tot}$
- 30 together with the theoretical  $S_{tot}$  associated with each retrieval.

#### 6.2 Sensitivity in the observation and retrieval spaces

Figure 5 shows the averaged observed and calculated (using a priori and retrievals) radiances together with the averaged calculated residuals for both B1 and B2. In B1, the mean residual is reduced from -0.8% (using the a priori) to 0.01% (using the retrievals) whereas in B2, the mean residual is reduced from -0.5% (using the a priori) to 0.01% (using the retrievals). The differences between the a priori residuals in B1 and B2 are due to the existence of more interfering parameters in B1 than in B2. Therefore, some differences between N<sub>2</sub>O\_B1 and N<sub>2</sub>O\_B2 due to the contamination of CH<sub>4</sub> and H<sub>2</sub>O are expected. Figures 6 and 7 show the mean N<sub>2</sub>O normalized (Deeter et al., 2007) averaging kernels matrix together with the altitude of the kernels maximum and the mean *CF* from CH<sub>4</sub>, temperature, surface temperature and H<sub>2</sub>O for N<sub>2</sub>O\_B1 and N<sub>2</sub>O\_B2, respectively. Considering the averaging kernels, the maximum of sensitivity is located at the retrieval level 309 hPa for both N<sub>2</sub>O B1 and

- N<sub>2</sub>O\_B2. In addition, the averaging kernels corresponding to this level peak at 309 hPa. Therefore, retrieved vmrs at this level are the most reliable for both N<sub>2</sub>O\_B1 and N<sub>2</sub>O\_B2. For N<sub>2</sub>O\_B2, all the averaging kernels peak at the levels 309 hPa. This means that the retrieved N<sub>2</sub>O vmr profiles are mainly sensitive to the real N<sub>2</sub>O vmr at this level. This result is consistent with previous studies from Kangah et al. (2017) and Xiong et al. (2014). The degree of freedom (DOF), which represents the number of independent vertical pieces of information of the retrieved profile and is computed as the trace of the averaging
- kernels matrix, is on average equal to 1.38 and 0.93 for N<sub>2</sub>O\_B1 and N<sub>2</sub>O\_B2, respectively. The DOF for N<sub>2</sub>O\_B1 is greater than that of N<sub>2</sub>O\_B2 because the SNR is higher in B1 than in B2. Thus, more channels with better SNR are selected in B1 than in B2. Although the retrieved N<sub>2</sub>O is impacted by temperature in the two bands, we have in B1 an additional significant impact of CH<sub>4</sub> and H<sub>2</sub>O. In conclusion, we expect more contamination on N<sub>2</sub>O\_B1 than on N<sub>2</sub>O\_B2.

#### 6.3 Retrieval accuracy

To assess the skills of the retrieval process, we applied the IASI  $N_2O$  averaging kernels to the HIPPO profiles using the following equation (Rodgers et al., 2000):

$$\hat{x} = Ax + (I - A)x_a \tag{10}$$

where  $x_a$  is the IASI a priori profile, x the HIPPO profile,  $\hat{x}$  the result of the averaging kernels application (called hereafter convolved HIPPO), I the identity matrix and A the IASI N<sub>2</sub>O averaging kernels matrix.

- Figures 8 and 9 show the results from the comparisons between HIPPO measurements and N<sub>2</sub>O\_B1 and N<sub>2</sub>O\_B2 averaged within the spatial and temporal window around the HIPPO measurements, respectively. N<sub>2</sub>O\_B1 and HIPPO measurements are moderately correlated (the Pearson linear correlation coefficient R=0.42) with a low bias and standard deviation (called hereafter std) error of -1.6 ppbv ( $\sim$ 0.5%) and 3.5 ppbv ( $\sim$ 1.0%), respectively. However, the quality of the retrievals depends on the latitude band. The consistency between N<sub>2</sub>O\_B1 and HIPPO increases at mid-latitudes (e.g. R=0.63 for northern hemi-
- sphere mid-latitudes). We can also notice that there is a very low mean bias (-0.1 ppbv) in the northern hemisphere high-latitude regions.