# Peer review of "IASI nitrous oxide $(N_2O)$ retrievals: validation and application to transport studies at daily time scales"

_Atmospheric Measurement Techniques, 2018_

## Referee Comment (RC1) · Anonymous Referee #1 · 5 May 2018

The paper deals with an application of IASI data to the retrieval of $N_2O$. The authors analyzed IASI data for the Northern-Hemisphere summer season (June-July) in 2011 and claim that the relatively high concentration of $N_2O$ over the Eastern Mediterranean basin is a result of pollution transport from Asia. I think that the paper shows potentially arguing and interesting results. However, an in-depth and accurate reading of the document shows, and I am sorry, that the paper presents a lot of weak points and many technical aspects, which need to be clarified and properly addressed before this study can be accepted for publication.

**General Comment**

It is known that the Mediterranean summer (June to September) is characterized by high pressure over the Mediterranean Europe and a low-pressure trough extending from the Persian Gulf through Iraq to the southeastern Mediterranean (see e.g., Y. Goldreich, Springer, 2003). It is now very well understood (e.g., Karnieli et al. JGR, 2009 and references therein) that this kind of weather pattern yields persistent northwesterly winds which causes long-range transport of air masses and pollutants from southeastern and southwestern Europe into the eastern Mediterranean basin. In agreement with this weather pattern, previous IASI studies (also cited in the present paper, e.g., doi:10.1364/OE.21.024753) have indeed evidenced higher concentrations of green-house gases over the Eastern Mediterranean basin. Conversely, the authors suggest that there could be another atmospheric pathway along which pollutants are transported to the Eastern Mediterranean basin. Because of the importance of this finding, the authors should be much more convincing in showing that their methodology has no weak points. In effect, their analysis is based on $N_2O$ profiles retrieved with less than 1 degree of freedom, and they concentrate on $N_2O$ layer average at $\approx309$ hPa, but they fail to show that this layer average has been independently resolved of the rest of the profile. In view of the broad structure of $N_2O$ AK they provide in the study, it is likely that they are mostly sensitive to the column amount of $N_2O$.

I have detailed my points below, which I hope can help authors to improve the paper.

**Major remarks**

1. Page 4, line 17. NEDT depends on the scene temperature, which, because of atmospheric absorption, is wave number dependent. Was this dependence taken into account? By the way, I suggest that the comparison should be made in the radiance space, using NEDN, which is wave-number independent.

2. Page 5, Equation (1). This equation should be introduced this way…*We used optimal estimation based on the Levenberg-Marquardt (put reference) algorithm as modified by Fletcher (put reference) and adapted for Optimal Estimation by Rodgers.* By the way, the important aspect here is that Equation (1), as it is written, is wrong. The term multiplying the leftmost $S_a^{-1}$ should be $(1 + \gamma)$ not simply, $\gamma$. In fact, Eq (1) should transform back to the OE estimator for $\gamma = 0$, which is not the case. Hope this is just a typo. Furthermore, how $\gamma$ is chosen at each step? Do authors perform retrieval in the BT space or radiance space? Please, clarify.

3. Page 5, Equation (2) is not consistent with Eq. (1). Apparently, the authors use $\gamma = 1$ for the final iterate, but then Equation (1) is not the correct OE estimator, and the final iterate would depart from optimality.

4. Page 5, Equation (3) applies just to one parameter. Considering that the authors claim to use a simultaneous approach, how is the a priori covariance of the whole state vector built up?

5. Page 5, line 20. Please show the $N_2O$ profile. The retrieval approach is strongly depending on such a background.

6. Page six, line 2. As before, show the $CO_2$ profile and those of other species used as a priori reference.

7. Page 6, line 13, what is a sink parameter? Please, explain.

8. Page 6, Equation (4). The denominator is wrong. The degrees of freedom of the $\chi^2$ form are dim($Y$). This can be demonstrated by a trivial use of the Standard Theorem of Least Squares (e.g., Rao 1973, the authors should consider that $\hat{X}$ is estimated from the data, so that the remaining degrees of freedom of data are dim($Y$)- dim($\hat{X}$) and dim($\hat{X}$) + [dim($Y$)- dim($\hat{X}$)] = dim($Y$)). Since the authors use a retrieval algorithm for which dim($\hat{X}$)≈dim($Y$), the $\chi^2$ is artificially decreased by a factor of almost 2.

9. Page 6, Equation (5) makes no sense in a Least Square retrieval approach, which seeks for a global minimum. Why one should look for a partial minimization, while using a simultaneous approach?

10. Page 7, lines 2 to 6. A $\chi^2$ variable with $n = dim(Y)$ degrees of freedom has mean $n$ and variance $2n$. Because for n large, the $\chi^2$ distribution is approximately Gaussian, to compute a $\chi^2_{th}$ −tolerance limit, say within 3 standard deviations (or $3\sigma$), we just need to calculate $\chi^2_{th} = n + 3\sqrt{2n}$. As an example, for $n = 103$, the number of channels the authors use in their B2 band, we have $\chi^2_{th}≈145$, or $\frac{\chi^2_{th}}{n} \approx 1.42$. Conversely, the authors use $\frac{\chi^2_{th}}{n} = 4$, which in view of the factor 2 above (see point 8) increase to 8, which corresponds to a tolerance interval of 50 (fifty) standard deviation (sic!). With this convergence criterion, almost all retrievals are not converged!

11. Page 7, Equation (6). I do not like the use of this empirical *Contamination Factor.* Why the final solution should be contaminated? They use a simultaneous retrieval. Averaging Kernels are good to assess vertical resolution. To check the interdependency of the retrieved state vector the authors have the a-posteriori covariance matrix. Please, use this matrix and compute the correlation matrix. In case the $N_2O$ profile has not been independently resolved, the authors will see a relatively large correlation with other parameters, e.g. $H_2O$. If so, they have only one way to go, change or improve the retrieval algorithm, e.g., by using more IASI channels, which are sensitive to $H_2O$ but not to $N_2O$. You have a lot in IASI.

12. Page 7 to 10, from Section Validation. With broad AK and a peak of 0.2 at most, the information retrieval comes from the background. There is no point in assessing this layer-by-layer accuracy. All AK broadly peaks around 400 mbar, which means that the $N_2O$ at this layer also receives contribution from the rest of profile. The authors need to check for correlation. In case, as I suspect, the $N_2O$ layer-retrieval at about 309 is strongly correlated with the rest of the profile, the accuracy alone (root mean square error of the a-posteriori covariance matrix) is not a good quality index. I suggest that the author should also compute the $N_2O$ total column. Because of integration along the vertical, this parameter will be more depending on the true state than the background.

13. Page 11, Section Troposphere variations….Since the authors failed to show that the $N_2O$ layer-retrieval at 309 hPa is independently resolved, the results in this section could be seriously flawed. Once again, by looking at AK in Fig. 6 and 7 the 309 hPa layer retrieval of $N_2O$ gets contribution from any other layer along the profile. It is important here that the authors show maps of the correlation matrix to get insight into a better understanding of the retrieval quality and accuracy at 309 hPa.

Minor Comments

14. Page, line 11. The authors say*, … Over the mid-latitude regions, both variations of N2O_B1 and N2O_B2 at 309 hPa are influenced by the stratospheric N2O-depleted air **because** of the relative coarse shape of the averaging kernel…*I have found clumsy sentences like this throughout the paper. It seems that the *cause* of a physical phenomenon is the mathematical structure of IASI AK. I think that authors here want to say that …*Because of the relative coarse shape of the averaging kernel, IASI is sensitive to variations of N2O_B1 and N2O_B2 at 309 hPa influenced by the stratospheric N2O-depleted air.* Here the authors seem to suggest that the best retrieval function capable of assessing this phenomenon could be a proper average over the broader part of AK, but then they go completely another way and focus on a single layer.

15. Page 3, section 2 IASI. Please put the reference to IASI at the beginning of the section….*IASI (Hilton et al 2011)*…Furthermore, on line 11 remove the reference to Clerbaux et al 2009, it is not appropriate once you have cited Hilton et al at very beginning. Furthermore, on line 13 the reference again to Hilton et al is redundant. Please, remove it.

*16.* Page 3, line 21, Change *RRTOV is a fast…* in *RTTOV is a polychromatic fast…*

17. Page 4, Beginning of section. That the IASI spectral coverage includes the $N_2O$ $\nu_1, \nu_3$ fundamental absorption band is a well assessed result from molecular spectroscopy, please rephrase the sentence on line 2, the reference to Clearbaux et al 2009 is not appropriate and unnecessary.

18. Table 1 is never called in the text. Please make proper reference to this table in the text and show also the number of IASI channels used for retrievals.

---

## Referee Comment (RC2) · Anonymous Referee #2 · 15 May 2018

This paper presents results from N2O IASI retrievals based on the RTTOV radiative transfer model. N2O satellite observations are important to understand its global distribution and maybe help characterizing its emissions. As mentioned below, IASI has already been used to retrieve N2O profiles and EUMETSAT retrieval algorithm is providing such data for the whole IASI period. Therefore the present study is not providing completely new data. It is nevertheless interesting to have more than one dataset from the same instrument inasmuch as the quality of the datasets are proven. The objective of the paper is two-folded. The presentation of the retrieval methodology and validation and a case study. As detailed below the methodology and validation part should largely be strengthened and the Hysplit transport study which is weak could be removed.

Overall the quality of this study is not good enough to be published in AMT. I have major concerns about the originality, the methodology and the results that are presented in the paper.

I-Originality of the work

There are other studies on IASI N2O retrievals which are not sufficiently acknowledged and discussed. One of the first publication about N2O IASI retrievals is Garcia et al. (AnnGeo, 2013). Based on one year of data they show that the N2O EUMETSAT v5 product (August et al., JQSRT, 2012) provides a good agreement with FTIR data at Izana for the 10-14 km vmr. Garcia et al. (AMT, 2016) make a comparison between the EUMETSAT v5 product and the Izana FTIR data for 4 years. These comparisons show a very good agreement (R=0.87) for the total columns annual cycle. In their latest paper Garcia et al. (AMTD, 2017) show a good agreement between IASI N2O and HIPPO data. The authors should use these previous studies in details rather than just citing them. In particular they should discuss and compare their retrieval methodology, characterization and results with those described in these papers throughout the manuscript.

IASI-A is flying since 2006 and the present paper presents retrievals for validation with HIPPO data and a series of situations over a limited region for a very limited time period. It is possible to accept such a limited study for a very recent mission but difficult for a ten years mission with previous studies much more extended already published. Indeed, as mentioned above, in their latest studies Garcia et al. have taken advantage of the long time series to make robust statistics and they have used different available validation datasets such as long term FTIR profiles and columns at Izana and GAW in-situ data (see reply to reviewers in Garcia et al., AMTD 2017) and HIPPO campaigns. The present paper would be more convincing if it could prove that the new IASI N2O retrievals provide robust information about the N2O variability taking advantage of the large IASI period which is not yet the case.

II-Methodology:

The retrieval methodology is not fully presented and justified. In page 5 the basic equation of the OEM are rewritten which is unnecessary. They are described and explained in Rodgers (2000) and many other publications and can therefor be removed and replaced by more interesting information. Indeed, the retrieval strategy itself is hardly described and justified. Many auxiliary parameters are retrieved together with the absorbing gases profiles but no justifications and no discussion about these retrieved parameters are given.

i-Contamination Factor: This part is interesting because it allows to document how uncertainty on an auxiliary parameter will impact the retrieved target state vector. Nevertheless it is only valid for auxiliary parameters that are kept constant and are not retrieved together with the target parameters. In case of retrieved parameters, it only gives an idea of the parameters which retrieval will mostly interfere with the target parameters but does not allow us to know the quantitative impact on the target parameters. The authors should explain that this methodology is not quantitative for retrieved parameters.

ii-Atmospheric temperature retrieval: Why is the atmospheric temperature retrieved together with N2O and the other interfering species? Where is the information about the atmospheric temperature profile coming from? Atmospheric temperature is normally retrieved from CO2 lines assuming constant CO2 vmr's. CO2 or other gases vmr's are retrieved assuming constant atmospheric temperatures. These procedures avoid mixing between T and gases retrievals. Here there are some CO2 lines in the B2 band but the most likely is that the temperature is retrieved from other absorption lines such as N2O. The risk of contamination and interference is therefore major. This is actually shown by figure 5 and 6 where the CF are drawn. Atmospheric temperature uncertainties have the largest impact on N2O retrievals in both B1 and B2 with CF a factor of 4 of more larger than for the other parameters in the mid-upper troposphere. As stated above, this means that the T and the N2O retrievals are not independent. Therefore

high (low) N2O could be caused by high (low) T or the other way around but the impact cannot be determined because of the joint retrieval.

iii-Emissivity retrieval: The authors state that in RTTOV the ocean emissivity is parameterized and that land emissivity is prescribed by an atlas. They call these emissivities a priori emissivities and retrieve surface emissivity in their procedure. Are the emissivities spectrally varying in RTTOV? How are the emissivity jacobians computed in RTTOV? Are they the same over sea and over land? It would be interesting to see results from emissivity retrievals and the differences over sea and land and over different types of land. Surface temperature and surface emissivity are parameters with signatures hard to discriminate in a small spectral window such as B1 or B2 as they basically give the background slope. The retrieval of both parameters is probably redundant. The authors should give information about how much the spectral chisquare has been improved when surface emissivity is retrieved and about the improvement it provides on the validation dataset. In case of no or too small improvements, the retrieval procedure has to be reconsidered without emissivity retrieval.

iv-Validation: Equation 10 is applied to the HIPPO profiles to take the IASI vertical resolution and the impact of the a priori profile into account. Nevertheless, in order to apply this equation, the validation profiles have to cover the whole atmosphere. How and with what data are the tropospheric HIPPO profiles completed above the aircraft profiles top? How is the tropopause altitude taken into account? Concerning the comparison between the empirical and the theoretical errors there is a conceptual error. The authors compare the standard deviations of the differences between smoothed validation profiles and retrieved profiles (Emp) to the theoretical error (sum of smoothing and measurement errors Theoret) (Fig. 4). But as the validation profiles are smoothed by equation 10, the smoothing error is already taken into account and Emp has to be compared to RetNoise. As RetNoise is larger than Smooth this would not make a big difference. The other way is to compute the differences between the retrieved profiles and the raw validation profiles and to compare Emp with Theoret. Furthermore, the

authors have shown that T uncertainty is largely impacting N2O retrievals (see CF) but as they retrieve jointly both parameters they cannot compute the resulting error. If the T profile was kept constant as suggested above, the errors caused T uncertainty could be evaluated (see Rodgers 2000). The errors caused by the other parameters should also be taken into account to compute the Theroret error but the same problem arises. The authors compute the Se matrix to provide the smoothing error instead of using Sa. Nevertheless Sa should represent the actual N2O global variability as accurately as possible and is the matrix that should be used to compute the smoothing error in equation 7 (Rodgers 2000). Se computed from the HIPPO data is representative of oceanic N2O profiles for given periods and may underestimate the variability. If the authors think it is a better representation of N2O global variability they have to justify this choice and may use it also for the retrievals. Furthermore a graphic representation of Se (diagonal values and covariance/correlation) should be given and compared to Sa.

Instead of R we should have r2 which shows the percentage variation in the retrieved profile that is explained by the variations of the validation profile. Therefore R> 0.707 is needed to have more than 50% of the retrieved variability coming from the real variability. It is also important to have a comparison of the variability of the validation data and of the retrieved data. All this information (standard deviation of the differences, r2, variability) should be given synthetically with a Taylor diagram.

III-Results:

i-Validation The retrieval results are not fully convincing. When the whole HIPPO dataset is used, meaning the strongest statistics (N about 100), r2=0.18 for B1 and 0.36 for B2 implying only 18 and 36% of the retrieved variability explained by the actual variability. Even if based on a limited HIPPO dataset, Garcia et al. (2017) achieve a better correlation (r2 = 0.58) whith a similar type of comparison as presented here. As they deal with a very close type of comparison, the results of Garcia et al. (2017) even in a paper under review should be discussed here. In most latitudinal bands (weaker

statistics with N < 30) r2 is lower than 0.5 especially in the B1 case with a maximum of 0.4 in the northern mid-latitudes. In the B2 case r2 is the highest (0.85) for the tropical southern latitudes. But in that case it is based on 12 points only which makes the statistics really poor and the high R is due to the fact that the points are separated in to clusters. Furthermore, in the best r2 cases (tropical southern and northern latitudes for B2) the slopes of the linear interpolation are much larger than unity (2.5 and 3.3) indicating a largely too strong variability of the retrieved vmr's compared to the validation vmr's. For northerm mid-latitudes r2 = 0.4 for B1 and 0.29 for B2 which are rather low values. Finally, the authors state that in summary N2O_B1 and B2 are of sufficient quality to analyse N2O variations in the mid and high latitude regions. This conclusion is not really supported by the validation results as discussed above. Especially for high northern latitudes with r2= 0.1 for both B1 and B2, only 10% of the variability comes from the actual N2O variability. We would rather say that these data should not be used.

ii-Transport study The variability of IASI N2O at 309 hPa shown on Fig. 13 is probably coming from a tropopause height difference. As shown by the AvK's, IASI vmr at 309 hPa is sensitive to a very large altitude range (600-120 hPa). Therefore it is equivalent to a N2O column or mean vmr over this range. When the tropopause changes from ∼100 hPa in the tropics to ∼250 hPa in the extratropics, the corresponding N2O columns mechanically change because the N2O vmr is lower in the stratosphere than in the troposphere. The authors attribute the N2O enhancement to upward transport from the Asian BL and horizontal transport within the anticyclone. This is also probably the case as shown by an extended literature based on satellite CO observations (Park et al., JGR, 2007...). Nevertheless, N2O is a well mixed gas and the quantification of such an effect is rather complicated. Surface in-situ data generally show a very limited seasonal variability of the N2O mixing ratio even in emission regions. Therefore the Asian BL is probably not N2O enriched as it is CO enriched. If the authors have evidence and data to document an important N2O enrichment during the monsoon in south Asia they should provide and discuss it. Another element that tends to

strengthen the tropopause effect is that the IASI N2O high values are not limited to the anticyclone boundaries but to the whole tropical region. See in particular the high N2O band between 15 and 5°N which is outside of the anticyclone (the southern boundary of the anticyclone is at about 15°N). In order to have a better idea of the tropopause versus BL transport effects (I) the region of Fig. 13 should be extended both in latitude and longitude (ii) the boundaries of the anticyclone should be provided on Fig. 13 based for instance on PV values (see Ploeger et al., ACP, 2017) or on geopotential height values (e.g. Randel and Park, JGR, 2006). The Hysplit study is based on online simulations and simply shows that on the southern edge of the anticyclone, transport is westward which is expected. It does not prove that the air parcels are coming recently from the south Asian BL (the backtrajectories end up between 700 and 300 hPa and with a tenths of trajectories the statistics are very poor when Lagrangian studies are performed with millions of air parcels) nor that N2O enhancements over the whole tropical band could be due to such a transport process. The Hysplit part is therefore largely insufficient to draw conclusions and could be removed. The literature is rich enough about the subject of upward transport of BL air masses to the UTLS and trapping of pollution into the anticyclone. See for instance the Lagrangian modeling study of Bergman et al. (2013). References to this extended litterature are enough.

IV-Minor comments:

p2l20-29: To my knowledge, the first paper to deal with tropospheric N2O retrievals from a satellite instrument is Chedin et al. (GRL, 2002). It shows very interesting results concerning the N2O evolution based on the TOVS instrument. This ref should be cited in the paper. p3l16: Turquety et al. (2004) does not concern IASI O3 retrievals. There are a number of recent refs concerning IASI O3 retrievals. P4l17-18: the authors should give a recent reference to justify their choice of NEDT. P5l16: the authors should give a ref or a detailed explanation that justify the shape of their a priori covariance matrix. We also need information about the shape of the a priori matrices for the other retrieved profiles (are they diagonal?). P6l2: the choice of 30% for the a priori

variability for H2O because of HDO is rather empirical and poorly justified. What does sink parameter mean? P6l4 and l6: sensitivity studies are mentioned but the reader knows nothing about what they are made of. Details about the methodology used and about the results of these sensitivity studies are needed. P6l14: ref for the radiometric noise (see above).

Figures: Fig 14: this figure is of poor quality and should be improved. The winds should be superimposed such as on Fig. 13 in order to make a more straightforward comparison.

---

## Author Comment (AC1) · 6 Aug 2018

The paper deals with an application of IASI data to the retrieval of $N_2O$. The authors analyzed IASI data for the Northern-Hemisphere summer season (June-July) in 2011 and claim that the relatively high concentration of $N_2O$ over the Eastern Mediterranean basin is a result of pollution transport from Asia. I think that the paper shows potentially arguing and interesting results. However, an in-depth and accurate reading of the document shows, and I am sorry, that the paper presents a lot of weak points and many technical aspects, which need to be clarified and properly addressed before this study can be accepted for publication.

**General Comment**

It is known that the Mediterranean summer (June to September) is characterized by high pressure over the Mediterranean Europe and a low-pressure trough extending from the Persian Gulf through Iraq to the southeastern Mediterranean (see e.g., Y. Goldreich, Springer, 2003). It is now very well understood (e.g., Karnieli et al. JGR, 2009 and references therein) that this kind of weather pattern yields persistent northwesterly winds which causes long-range transport of air masses and pollutants from southeastern and southwestern Europe into the eastern Mediterranean basin. In agreement with this weather pattern, previous IASI studies (also cited in the present paper, e.g., doi:10.1364/OE.21.024753) have indeed evidenced higher concentrations of green-house gases over the Eastern Mediterranean basin. Conversely, the authors suggest that there could be another atmospheric pathway along which pollutants are transported to the Eastern Mediterranean basin. Because of the importance of this finding, the authors should be much more convincing in showing that their methodology has no weak points. In effect, their analysis is based on $N_2O$ profiles retrieved with less than 1 degree of freedom, and they concentrate on $N_2O$ layer average at ~309 hPa, but they fail to show that this layer average has been independently resolved of the rest of the profile. In view of the broad structure of $N_2O$ AK they provide in the study, it is likely that they are mostly sensitive to the column amount of $N_2O$.

$\rightarrow$ The transport process between the Asian surface and the eastern Mediterranean during summer monsoon has been already demonstrated in Kangah et al., (2017) using GOSAT $N_2O$ retrievals, LMDz-Or-INCA chemical transport model and backtrajectories together with an extended literature. The aim of this paper is not to prove again this finding but to validate new IASI $N_2O$ retrievals showing that these retrievals can capture this transport process at daily time scale (part 7 of the manuscript). Concerning the vertical resolution of the data refer to response #12 and #14.

I have detailed my points below, which I hope can help authors to improve the paper.

**Major remarks**

1. Page 4, line 17. NEDT depends on the scene temperature, which, because of atmospheric absorption, is wave number dependent. Was this dependence taken into account? By the way, I suggest that the comparison should be made in the radiance space, using NEDN, which is wave-number independent.

Spectral NEDT at 280 K is the reference value for IASI radiometric noise and is often used to check sensitivity of any retrieval to the measurement. Therefore, in our study, the NEDT is given at a reference temperature of 280 K. The incriminated sentence has been modified into:

> The IASI radiometric noise expressed as the Noise Equivalent Delta Temperature (NEDT) at the reference temperature of 280 K is superimposed to the $|\Delta BT|$ signals.

For NEDN, we do not agree with the statement of the reviewer saying that it is wavenumber independent. Considering Figure 1 from Amato et al. (1995), then it is obvious that NEDN is varying with wavenumber (see Figure R1).

[Figure]

**Figure R1**: *IASI radiometric noise as a function of Wavenumber (taken from Amato et al., 1995).*

Amato, U., Cuomo, V., and Serio, C.: Assessing the impact of radiometric noise on IASI performances, Remote Sensing, 16(15), 2927-2938, 1995.

In addition, the most important here is the use of the same unit to compare sensitivity of the forward model to the geophysical parameters relative to the IASI radiometric noises.

2. Page 5, Equation (1). This equation should be introduced this way... *We used optimal estimation based on the Levenberg-Marquardt (put reference) algorithm as modified by Fletcher (put reference) and adapted for Optimal Estimation by Rodgers.* By the way, the important aspect here is that Equation (1), as it is written, is wrong. The term multiplying the leftmost $S_a^{-1}$ should be $(1 + \gamma)$ not simply, $\gamma$. In fact, Eq (1) should transform back to the OE estimator for $\gamma = 0$, which is not the case. Hope this is just a typo. Furthermore, how $\gamma$ is chosen at each step? Do authors perform retrieval in the BT space or radiance space? Please, clarify.

$\rightarrow$ We verified the equation (1) and the reviewer is right, it is a typo. It should be $(1 + \gamma)$. We modified the equation accordingly.

$$\hat{X}_{i+1} = X_a + \left(K_i^T S_y^{-1} K_i + (1 + \gamma) S_a^{-1}\right)^{-1} \times \left\{K_i^T S_y^{-1}\left(\left[Y - F(\hat{X}_i)\right] + K_i\left[\hat{X}_i - X_a\right]\right) + \gamma S_a^{-1}\left[\hat{X}_i - X_a\right]\right\} \quad (1)$$

We have modified the introduction of the equation (1) into:

> We used optimal estimation based on the Levenberg-Marquardt (Levenberg, 1944; Marquardt, 1963) algorithm and adapted for Optimal Estimation by Rodgers (Rodgers, 2000)…

Note that we did not used the Fletcher strategy.
We have inserted the 2 following references in the revised version:

> Levenberg, K.: A method for the solution of certain nonlinear
>     problems in least squares, Quart. Appl. Math., 2, 164, 1944.
> Marquardt, D. W.: An algorithm for least-squares estimation
>     of nonlinear parameters, SIAM J. Appl. Math., 11, 431, 1963.

→The initial $\gamma$ is initialised to 10. At each step, $\gamma$ is updated as follows:

> ➢ If the cost function $\chi^2$ (cf. response #8) decreases: $\gamma$ is divided by 5 for the next step
> ➢ If the cost function $\chi^2$ increases: $\gamma$ is multiplied by 5 and both the cost function and the estimated profile ($\hat{X}_{i+1}$) are recalculated.

3. Page 5, Equation (2) is not consistent with Eq. (1). Apparently, the authors use $\gamma$ = 1 for the final iterate, but then Equation (1) is not the correct OE estimator, and the final iterate would depart from optimality.

→ With the equation (1) correctly written, we have clarified the value of $\gamma$ (= 0) for the final iteration into:

> The vertical sensitivity of the retrieval can be characterised
> using the averaging kernel matrix (A) defined as (with $\gamma$ = 0
> for the final iteration):

4. Page 5, Equation (3) applies just to one parameter. Considering that the authors claim to use a simultaneous approach, how is the a priori covariance of the whole state vector built up?

→ The parameters are independent to each other in building up $S_a$ thus the extra "block-diagonal" elements of $S_a$ are fixed to 0 (there are no apriori correlation errors between the different state vector parameters). We have clarified this point by inserting the following sentence:

…The a priori error covariance matrix $S_a$ is built for all chemical
species and by considering parameters independent to each other as
follows (cf. Rodgers, 2000):

$$S_{aij} = \sigma_a^2 \times \exp(-|\ln(P_i) - \ln(P_j)|) \tag{3}$$

where $\sigma_a^2$ is an a priori variance error fixed for each parameter of
the state vector and $P_i$ the pressure level at the level $i$.

Diagonal matrices are used for temperature profile and surface emissivity…

5. Page 5, line 20. Please show the N2O profile. The retrieval approach is strongly depending on such a background.

→ Done (cf. Figure 3)

6. Page six, line 2. As before, show the $CO_2$ profile and those of other species used as a priori reference.

→ Done (cf. Figure 3)

7. Page 6, line 13, what is a sink parameter? Please, explain.

→ We removed part of the sentence linked to the term "sink parameter" that is too much confusing.

8. Page 6, Equation (4). The denominator is wrong. The degrees of freedom of the $\chi^2$ form are dim(Y). This can be demonstrated by a trivial use of the Standard Theorem of Least Squares (e.g., Rao 1973, the authors should consider that $\hat{X}$ is estimated from the data, so that the remaining degrees of freedom of data are dim($Y$)-dim($\hat{X}$) and dim($\hat{X}$)+[dim(Y)-dim($\hat{X}$)]=dim($Y$)). Since the authors use a retrieval algorithm for which dim($\hat{X}$)~dim(Y), the $\chi^2$ is artificially decreased by a factor of almost 2.

→ Equation (4) refers to the normalized cost function $\chi^2_{\text{norm}}$ and not to the $\chi^2$ test. We have rewritten the sentence in order to define (1) the cost function $\chi^2$ and (2) the normalized cost function $\chi^2_{\text{norm}}$.

Our retrieval process consists in the minimization of the cost function $\chi^2$ defined as:

$$\chi^2 = \left[\hat{X} - X_a\right]^T S_a^{-1}\left[\hat{X} - X_a\right] + \left[Y - F(\hat{X})\right]^T S_y^{-1}\left[Y - F(\hat{X})\right] \tag{4}$$

We used the normalized cost function $\chi^2_{\text{norm}}$ to evaluate the quality of the retrieval by combining the calculated residuals relative to the observations error covariance matrix and the difference between the estimated and the a priori profiles relative to the a priori error covariance matrix:

$$\chi^2_{\text{norm}} = \frac{[\hat{X}-X_a]^T S_a^{-1}[\hat{X}-X_a]+[Y-F(\hat{X})]^T S_y^{-1}[Y-F(\hat{X})]}{\text{dim}(\hat{X})+\text{dim}(Y)} \tag{5}$$

where $\text{dim}(\hat{X})$ and $\text{dim}(Y)$ are the dimensions of the state vector and of the radiances (number of channels), respectively. In theory, $\chi^2_{\text{norm}}$ should be close to unity.

Figure R2 shows histograms of $\chi^2_{\text{norm}}$ for converged pixels of N$_2$O_B1 and N$_2$O_B2 over the region corresponding to figures 14 and 15 of the manuscript. $\chi^2_{\text{norm}}$ is globally higher for

N₂O_B1 than for N₂O_B2 confirming the difficulties to minimize the cost function in B1 compared to B2.

[Figure]

**Figure R2**: $\chi^2_{\mathrm{norm}}$ histograms for N₂O_B1 (left) and N₂O_B2(right).

9. Page 6, Equation (5) makes no sense in a Least Square retrieval approach, which seeks for a global minimum. Why one should look for a partial minimization, while using a simultaneous approach?

→ Equation (5) is not a partial minimization but another variable to assess the quality of the N₂O retrievals. In order to suppress the ambiguity with the cost function, we have redefined this variable Q_N2O for "quality of the N2O retrievals". We have modified the incriminated sentences accordingly. This parameter is a kind of normalized difference between The retrieval and the apriori N2O profile and is used to reject unrealistic N2O retrievals.

In addition to $\chi^2_{\mathrm{norm}}$, we computed another variable, $Q_{\mathrm{N_2O}}$, to assess the quality of the retrieved tropospheric N₂O profile which is our target species defined as the difference between the a priori and the retrieved N₂O relative to the N₂O a priori errors $\sigma_a$:

$$Q_{\mathrm{N_2O}} = \sum_{P_j<1000\,\mathrm{hPa}}^{P_j>200\,\mathrm{hPa}}[\hat{X}_j - X_{aj}]^2 \beta_{aj}/n_p \qquad (6)$$

where $\hat{X}_j$ and $X_{aj}$ are the retrieved parameter and the a priori N₂O at the pressure level $P_j$, respectively. $\beta_{aj}$ is the diagonal element of the a priori error precision matrix (the inverse of the a priori error covariance matrix) at the pressure level $P_j$ and $n_p$ is the number of levels used for the calculation.

An upper limit $\chi^2_{\mathrm{threshold}}$ for the $\chi^2_{\mathrm{norm}}$ parameter is generally used to select good quality pixels. For instance, $\chi^2_{\mathrm{threshold}} = 3$ on a $\chi^2_{\mathrm{norm}}$ calculated in the radiances space was used to select good quality pixels for CH₄ retrievals from IASI measurements (Xiong et al., 2013). Following the same methodology, we applied an upper limit $Q_{\mathrm{N_2O}}^{\mathrm{threshold}}$ on $Q_{\mathrm{N_2O}}$ to select good quality pixels. After performing sensitivity studies for both N₂O_B1 and N₂O_B2, we selected all the IASI pixels with $\chi^2_{\mathrm{norm}} \leq$

$\chi^2_{\text{threshold}}$ and $Q_{\text{N}_2\text{O}} \leq Q^{\text{threshold}}_{\text{N}_2\text{O}}$ with $\chi^2_{\text{threshold}} = 4$ and $Q^{\text{threshold}}_{\text{N}_2\text{O}} = 4$.

10. Page 7, lines 2 to 6. A $\chi^2$ variable with $n = dim(Y)$ degrees of freedom has mean $n$ and variance $2n$. Because for n large, the $\chi^2$ distribution is approximately Gaussian, to compute a $\chi^2_{th}$ –tolerance limit, say within 3 standard deviations (or $3\sigma$), we just need to calculate $\chi^2_{th} = n + 3\sqrt{2n}$. As an example, for $n = 103$, the number of channels the authors use in their B2 band, we have $\chi^2_{th} \sim 145$, or $\frac{\chi^2_{th}}{n} \sim 1.42$. Conversely, the authors use $\frac{\chi^2_{th}}{n} = 4$, which in view of the factor 2 above (see point 8) increase to 8, which corresponds to a tolerance interval of 50 (fifty) standard deviation (sic!). With this convergence criterion, almost all retrievals are not converged!

$\rightarrow$ As we previously explained (cf. #8 and #9), the parameter $\chi^2$ is the cost function to be minimised and $\chi^2_{\text{norm}}$ is used as a quality parameter and not as a convergence criterion. However, the convergence criterion is performed by computing another parameter ($d_2$) which is roughly the distance between the next and the previous values of the forward model relative to the measurement errors covariance matrix:

$d_2 = [Y_{i+1} - Y_i]^T S_y^{-1} [Y_{i+1} - Y_i]$

Thus, we converge when $d_2$ is lower than the dimension of $Y$ (namely the number of channels).

11. Page 7, Equation (6). I do not like the use of this empirical *Contamination Factor*. Why the final solution should be contaminated? They use a simultaneous retrieval. Averaging Kernels are good to assess vertical resolution. To check the interdependency of the retrieved state vector the authors have the a-posteriori covariance matrix. Please, use this matrix and compute the correlation matrix. In case the $N_2O$ profile has not been independently resolved, the authors will see a relatively large correlation with other parameters, e.g. $H_2O$. If so, they have only one way to go, change or improve the retrieval algorithm, e.g., by using more IASI channels, which are sensitive to $H_2O$ but not to $N_2O$. You have a lot in IASI.

$\rightarrow$ First, we have to clarify the fact that the Contamination Factor is mainly derived from the averaging kernel of the whole state vector. According to the definition of the averaging kernel matrix, the block-diagonal matrix of A represents the averaging kernel matrix of each retrieved parameters and the extra block matrices represent the interference matrices between the different co-retrieved parameters. As it was demonstrated by Rodgers and Connors (2003), these interference matrices are sources of "interference errors" on the target species. Thus, we modified the text accordingly:

```
the Contamination Factor (called hereafter CF) defined as
follows:
the Contamination Factor (called hereafter CF) defined as
follows:
```

$$CF(i) = \sum_j \left| A_{xc_{(ij)}} \right| \frac{\Delta c_j}{x_i} \times 100 \qquad (7)$$

Where $A_{xc_{(ij)}} = \frac{\partial \hat{x}_i}{\partial c_j}$ is the submatrix of A corresponding to the interference between the co-retrieve parameter c and the target retrieved species x (Rodgers and Connor, 2003); $\Delta c_j$ is the uncertainty on the parameter c at the level j and; $CF(i)$ is the contamination of the parameter c on the retrieved $N_2O$ $\hat{x}_i$ at the level $i$.

We have also inserted the following reference in the revised version:

Rodgers, C. D., and B. J. Connor (2003), Intercomparison of remote sounding instruments, J. Geophys. Res., 108, 4116, doi: 10.1029/2002JD002299

In addition, according to the definition of the averaging kernel matrix (cf. eq (2) of the manuscript) there is the following link between *A* and the a-posteriori errors covariance matrix $S_x$:

$A = S_x K^T S_y^{-1} K = I - S_x S_a^{-1}$

Thus, the interference matrices of *A* give almost the same information as the extra-diagonal submatrices of the a-posteriori errors covariance matrix $S_x$.

Concerning the possibility of using more channels sensitive to $H_2O$, the characterisation of the $H_2O$ itself is not the problem since we have as apriori knowledge the operational IASI level 2 and the corresponding error variance. The difficulty here is to characterise the different spectral line comb of the isotopic component of $H_2O$ (especially HDO) and to remove as far as possible the induced contamination of the $N_2O$ profile. So, adding more channels is not the solution. Figure R2 show the isotopic ratio multiply by the concentration and multiply by the cross section of PAN, HDO and HNO3 in B1. The H2O vmr is from IASI operational level 2 product and the concentration of PAN and HNO3 are respectively from Fischer et al. (2014) and Wespes et al., (2007) respectively. The spectroscopic database used here is from HITRAN. This figure shows an important impact of these component in B1. However, the most important component between these three is HDO as it was shown in Clerbaux et al., (2009) since PAN is highly variable and HNO3 has a low impact on the transmittance. In addition, Liuzzi et al. (2016) shows the impact of HDO in B1 by analysing the IASI spectral residuals (obs-calc) with and without retrieving HDO. Thus, we modify the manuscript as follows:

…Clerbaux et al. (2009) and Liuzzi et al. (2016) show the presence of a relatively strong absorption band of the deuterium hydrogen oxide (HDO) also called semiheavy water in the band B1…

We inserted the following reference in the revised version:

Liuzzi, G., Masiello, G., Serio, C., Venafra, S., & Camy-Peyret, C.: Physical inversion of the full IASI spectra: Assessment of atmospheric parameters retrievals, consistency of spectroscopy and forward modelling. Journal of Quantitative Spectroscopy and Radiative Transfer, 182, 128-157, https://doi.org/10.1016/j.jqsrt.2016.05.022, 2016.

[Figure]

***Figure R3****: Spectral variations of the isotopic ratio multiply by the concentration and multiply by the cross section of PAN, HDO and HNO3 in B1.*

12. Page 7 to 10, from Section Validation. With broad AK and a peak of 0.2 at most, the information retrieval comes from the background. There is no point in assessing this layer-by-layer accuracy. All AK broadly peaks around 400 mbar, which means that the $N_2O$ at this layer also receives contribution from the rest of profile. The authors need to check for correlation. In case, as I suspect, the $N_2O$ layer- retrieval at about 309 is strongly correlated with the rest of the profile, the accuracy alone (root mean square error of the a-posteriori covariance matrix) is not a good quality index. I suggest that the author should also compute the $N_2O$ total column. Because of integration along the vertical, this parameter will be more depending on the true state than the background.

→ We agree with the reviewer concerning the fact that the $N_2O$ retrieval at 309 hPa is impacted by the apriori background since the peak is around 0.2. However, the apriori profile is fixed so the variations observed in the retrievals at 309 hPa are not due to the apriori background.
Figure 7 and 8 show that the peak of the averaged *A* is around 0.22 at 309 hPa and around 0.12 at 309 hPa for $N_2O\_B1$ and $N_2O\_B2$ respectively.   These results are consistent with the previous studies from Kangah et al. (2017) and Garcia et al. (2018). The only way to figure out the relevance of using the dominant layer (309 hPa) is through validation studies including

large and representative reference in-situ datasets and scientific analyses of the strengths and weaknesses of these one-layer retrievals. Thus, a retrieval can be considered either as an estimation of the reality or as an estimation of a smoothed reality (Rodgers, 2000). Here we chose to consider the retrievals as estimations of the reality and then analysed the smoothing effects.

So, our work is also to show that despite the smoothing effect due to the shape of the averaging kernel, the retrieval at 309 hPa is sufficiently representative of this layer to study transport processes.

Kangah et al. (2017) studies upper tropospheric transport processes between Asia and the eastern Mediterranean using GOSAT/TANSO $N_2O$ retrieval at 314 hPa. GOSAT averaging kernels have almost the same shape as our IASI $N_2O$ averaging kernel, peaking at around 300 hPa with a full width at half maximum from ~500 hPa to 100 hPa. Despite the broad shape of $A$ and a peak hardly higher than 0.1, this study shows that GOSAT $N_2O$ retrievals at 314 hPa allow to study upper tropospheric $N_2O$ transport processes between Asia and the eastern Mediterranean at monthly timescale. In the present paper, we show that our retrieval at 309 hPa capture these upper tropospheric transport processes at daily timescale.

In addition, refer to response #I to the referee #2 about the added value of this work compared with the previous studies.

13. Page 11, Section Troposphere variations... Since the authors failed to show that the $N_2O$ layer- retrieval at 309 hPa is independently resolved, the results in this section could be seriously flawed. Once again, by looking at AK in Fig. 6 and 7 the 309 hPa layer retrieval of $N_2O$ gets contribution from any other layer along the profile. It is important here that the authors show maps of the correlation matrix to get insight into a better understanding of the retrieval quality and accuracy at 309 hPa.

→ Cf. #12 and #11

**Minor Comments**

14. Page 1, line 11. The authors say, ... *Over the mid-latitude regions, both variations of N2O_B1 and N2O_B2 at 309 hPa are influenced by the stratospheric N2O-depleted air* **because** *of the relative coarse shape of the averaging kernel...*I have found clumsy sentences like this throughout the paper. It seems that the cause of a physical phenomenon is the mathematical structure of IASI AK. I think that authors here want to say that... *Because of the relative coarse shape of the averaging kernel, IASI is sensitive to variations of N2O_B1 and N2O_B2 at 309 hPa influenced by the stratospheric N2O-depleted air.* Here the authors seem to suggest that the best retrieval function capable of assessing this phenomenon could be a proper average over the broader part of AK, but then they go completely another way and focus on a single layer.

→ N2O_B1 and N2O_B2 variations are not only due to physical phenomena.
They contain both physical structures from transport processes and other structures partly due to the smoothing effect of the averaging kernel matrix at this level. The physical structure has been clearly demonstrated throughout the paper through validation with in-situ datasets and by showing transport processes at 309 hPa consistent with an extended literature (see

Kangah et al., 2017). We also want to show here the weaknesses of the retrievals at 309 hPa by analysing the impact of these mathematical smoothing effects on N2O_B1 and N2O_B2 variations.

15. Page 3, section 2 IASI. Please put the reference to IASI at the beginning of the section... *IASI (Hilton et al., 2011)...* Furthermore, on line 11 remove the reference to Clerbaux et al. (2009), it is not appropriate once you have cited Hilton et al. at very beginning. Furthermore, on line 13 the reference again to Hilton et al is redundant. Please, remove it.

$\rightarrow$ Done

16. Page 3, line 21, Change *RRTOV is a fast...* in *RTTOV is a polychromatic fast...*

$\rightarrow$ Done

17. Page 4, Beginning of section. That the IASI spectral coverage includes the $N_2O$ $\nu 1, \nu 3$ fundamental absorption band is a well assessed result from molecular spectroscopy, please rephrase the sentence on line 2, the reference to Clerbaux et al. (2009) is not appropriate and unnecessary.

$\rightarrow$ We have rephrased the sentence into:

> From molecular spectroscopy (Rothman et al., 2009), it is known that three absorption bands of $N_2O$ are present in the IASI spectral range centred at $\sim$1280 cm$^{-1}$, $\sim$2220 cm$^{-1}$ and $\sim$2550 cm$^{-1}$.

> Rothman, L. S., Gordon, I. E., Barbe, A., Benner, D. C., Bernath, P. F., Birk, M., et al., The HITRAN 2008 molecular spectroscopic database, Journal of Quantitative Spectroscopy and Radiative Transfer, 110(9-10), 533-572, 2009.

18. Table 1 is never called in the text. Please make proper reference to this table in the text and show also the number of IASI channels used for retrievals.

$\rightarrow$ We have inserted the following sentence in the section 5.1.

> Table 1 synthetises the a priori standard deviation errors ($\sigma_a$) used for each retrieved parameter in B1 and B2.

$\rightarrow$ Although the number of IASI channels selected for the retrievals is written in section 4 and Figure 2, we have clarified this point by inserting a new sentence at the end of section 4.

> To sum up, the number of IASI channels used for the retrievals in the bands B1, B2 and B3 is $N$ = 126, 103 and 0, respectively.

---

## Author Comment (AC2) · 6 Aug 2018

This paper presents results from N2O IASI retrievals based on the RTTOV radiative transfer model. N2O satellite observations are important to understand its global distribution and maybe help characterizing its emissions. As mentioned below, IASI has already been used to retrieve N2O profiles and EUMETSAT retrieval algorithm is providing such data for the whole IASI period. Therefore, the present study is not providing completely new data. It is nevertheless interesting to have more than one dataset from the same instrument in as much as the quality of the datasets are proven. The objective of the paper is two-folded. The presentation of the retrieval methodology and validation and a case study. As detailed below the methodology and validation part should largely be strengthened and the Hysplit transport study which is weak could be removed.

Overall the quality of this study is not good enough to be published in AMT. I have major concerns about the originality, the methodology and the results that are presented in the paper.

I-Originality of the work

There are other studies on IASI N2O retrievals which are not sufficiently acknowledged and discussed. One of the first publication about N2O IASI retrievals is Garcia et al. (AnnGeo, 2013). Based on one year of data they show that the N2O EUMETSAT v5 product (August et al., JQSRT, 2012) provides a good agreement with FTIR data at Izana for the 10-14 km vmr. Garcia et al. (AMT, 2016) make a comparison between the EUMETSAT v5 product and the Izana FTIR data for 4 years. These comparisons show a very good agreement (R=0.87) for the total columns annual cycle. In their latest paper Garcia et al. (AMTD, 2017) show a good agreement between IASI N2O and HIPPO data. The authors should use these previous studies in details rather than just citing them. In particular they should discuss and compare their retrieval methodology, characterization and results with those described in these papers throughout the manuscript.

$\rightarrow$ Interannual trends and seasonality of $N_2O$ have been widely addressed by the retrieved data presented in the previous studies. This work aims to present a new retrieved $N_2O$ data which are of a sufficient quality to be used to study spatial and temporal variation of $N_2O$ on a daily basis. As it was demonstrated in Kangah et al., (2017), these variations could be footprints of high $N_2O$ emission hotspots especially over Asia. Thus, to enhance the originality of this work in comparisons to the other studies, we added some sentences in the introduction part of the paper as follows:

> These operational N2O total column products also show
> seasonal cycles and annual trends consistent with the
> retrieved N2O from the ground-based Fourier Transform
> Spectrometer (FTS) observations at the Izaña Atmospheric
> Observatory (IZO, Spain) (García et al., 2014, 2016). First
> results of N2O total columns retrievals using a partially

scanned IASI interferogram with an accuracy of ±13 ppbv (~4%) are described in Grieco et al. (2013). Retrievals of N2O tropospheric profiles have been performed using the Atmospheric Infrared Sounder (AIRS) and the results showed global interannual trends consistent with surface measurements (Xiong et al., 2014). N2O profiles retrieved from the Greenhouse Gas Observing Satellite (GOSAT) measurements have been used to study the transport of Asian summertime high N2O emissions to the Mediterranean upper troposphere (Kangah et al., 2017). Using monthly averaged GOSAT N2O retrievals at 314 hPa together with outputs from the chemistry transport model LMDz-Or-INCA, this study evidenced the transport of high surface N2O emissions from Asia to the upper tropospheric Mediterranean during the summer monsoon period. This was the first study to link upper tropospheric N2O spatial and temporal variations to regional emissions hotspots seasonality using retrievals from satellite measurements. In this paper, we describe the IASI instrument and the Radiative Transfer for Tiros Operational Vertical sounder (RTTOV) used as forward model in our retrieval system in sections 2 and 3, respectively. We present the retrieval strategy and the validation of the results using HIPPO airborne in situ measurements in sections 5 and 6, respectively. In section 7, we analyse the scientific consistency of the retrievals focusing on the long-range transport of N2O during the Asian summer Monsoon. In this part, we show that the $N_2O$ transport processes between the Asian surface and the eastern Mediterranean demonstrated in kangah et al., (2017) can be followed using our retrievals at a finer timescale, namely on a daily basis.

Moreover, the consistency between our retrieval vertical sensitivity and the previous studies is detailed in the manuscript as follows:

In addition, the averaging kernels corresponding to this level peak at 309 hPa. Therefore, retrieved vmrs at this level are the most reliable for both N2O_B1 and N2O_B2. For N2O_B2, all the averaging kernels peak at the levels 309 hPa. This means that the retrieved N2O vmr profiles are mainly sensitive to the real

N2O vmr at this level. This result is consistent with previous studies from Kangah et al. (2017), Xiong et al. (2014), Grieco et al. (2013) and Garcia et al. (AMT, 2018).

The following reference have been inserted in the revised version:

García, O. E., Schneider, M., Ertl, B., Sepúlveda, E., Borger, C., Diekmann, C., Wiegele, A., Hase, F., Barthlott, S., Blumenstock, T., Raffalski, U., Gómez-Peláez, A., Steinbacher, M., Ries, L., and de Frutos, A. M.: The MUSICA IASI CH4 and N2O products and their comparison to HIPPO, GAW and NDACC FTIR references, Atmos. Meas. Tech., 11, 4171-4215, https://doi.org/10.5194/amt-11-4171-2018, 2018.

Concerning the retrieval accuracy, we inserted the following paragraph in the revised version:

… MUSICA/IASI retrieved $N_2O$ (Garcia et al., 2018) presented a R2 (0.22) nearly similar to that of N2O_B1 (0.18) but with a greater std error (~2.5 %). Comparing with HIPPO, GOSAT $N_2O$ retrievals (Kangah et al., 2017) have a std error of about 0.6 %, a R2 of about 0.19 and a slope of about 0.22. In addition, MUSICA/IASI retrieved $N_2O$ have a degree of fredom of about 1.39 which is nearly the same than the one of N2O_B1 (1.38).
 These results must be compared very carefully with those of the present paper since there are significant differences in the error analysis strategy and reference datasets between the different studies. Thus, in the one hand, MUSICA/IASI retrieved $N_2O$ are compared with HIPPO datasets using a smaller number of collocated pixels (N=23) than that we used (98 and 102 collocated data for N2O_B1 and N2O_B2, respectively) and in the other hand, GOSAT $N_2O$ retrievals have been validated only over maritime pixels. Since the linear regression is very sensitive to this kind of differences, we can only assess a very qualitative and approximative comparison between these retrievals. Therefore, we can consider, at first glance, that MUSICA/IASI retrieved $N_2O$ seems qualitatively consistent with N2O_B1 and GOSAT $N_2O$ in terms of accuracy and vertical sensitivity.

To assess an exact and detailed comparison between the different kind of retrieved $N_2O$, an inter-comparison study must be performed. This is out of scope of this paper which aims to present a new retrieved $N_2O$ results with their strength and weaknesses and the kind of scientific study these new $N_2O$ products can be used for.

IASI-A is flying since 2006 and the present paper presents retrievals for validation with HIPPO data and a series of situations over a limited region for a very limited time period. It is possible to accept such a limited study for a very recent mission but difficult for a ten years mission with previous studies much more extended already published. Indeed, as mentioned above, in their latest studies Garcia et al. have taken advantage of the long time series to make robust statistics and they have used different available validation datasets such as long term FTIR profiles and columns at Izana and GAW in-situ data (see reply to reviewers in Garcia et al.,

AMTD 2017) and HIPPO campaigns. The present paper would be more convincing if it could prove that the new IASI N2O retrievals provide robust information about the N2O variability taking advantage of the large IASI period which is not yet the case.

$\rightarrow$ The HIPPO campaigns cover the 4 seasons and almost all the latitudinal bands (cf. figure 4 of the manuscript). There are therefore, at this time the best database to validate $N_2O$ retrievals. In addition, our aim (cf. response #I-) is to demonstrate that our data are useful to study spatial and temporal variations of $N_2O$ on a daily basis. Our strategy here is to take advantage of the knowledge of the summertime $N_2O$ transport processes between the Asian surface and the Mediterranean which have been assessed on a monthly basis in the previous study of Kangah et al., (2017) and show that our retrievals can capture this transport with finer timescales.  For this purpose, we don't need to use all IASI data since 2006.

**II-Methodology:**

The retrieval methodology is not fully presented and justified. In page 5 the basic equation of the OEM are rewritten which is unnecessary. They are described and explained in Rodgers (2000) and many other publications and can therefor be removed and replaced by more interesting information. Indeed, the retrieval strategy itself is hardly described and justified. Many auxiliary parameters are retrieved together with the absorbing gases profiles but no justifications and no discussion about these retrieved parameters are given.

$\rightarrow$ We clarified the retrieval methodology in the revised version of the manuscript (cf. responses #2, #3, #4, #8 to the referee #1). Concerning, the auxiliary parameters, the figure 2 of the manuscript highlights the key parameters in each band which should be accurately known to perform good $N_2O$ retrievals. In addition, error covariance matrix used for these by-products are also described and justified.  Furthermore, the last important thing about these parameters is to estimate and as far as possible to remove their impact on the $N_2O$ retrievals. This is what we did by using the Contamination Factor (see #II-i).

i-Contamination Factor: This part is interesting because it allows to document how uncertainty on an auxiliary parameter will impact the retrieved target state vector. Nevertheless it is only valid for auxiliary parameters that are kept constant and are not retrieved together with the target parameters. In case of retrieved parameters, it only gives an idea of the parameters which retrieval will mostly interfere with the target parameters but does not allow us to know the quantitative impact on the target parameters. The authors should explain that this methodology is not quantitative for retrieved parameters.

$\rightarrow$ We do not agree with the referee concerning the Contamination. In a simultaneous retrieval strategy, we can also assess how uncertainty on a co-retrieved parameter will impact the target species (see response #11 to the referee #1).

ii-Atmospheric temperature retrieval: Why is the atmospheric temperature retrieved together with N2O and the other interfering species? Where is the information about the atmospheric temperature profile coming from? Atmospheric temperature is normally retrieved from CO2 lines assuming constant CO2 vmr's. CO2 or other gases vmr's are retrieved assuming constant atmospheric temperatures. These procedures avoid mixing between T and gases retrievals.

Here there are some CO2 lines in the B2 band but the most likely is that the temperature is retrieved from other absorption lines such as N2O. The risk of contamination and interference is therefore major. This is actually shown by figure 5 and 6 where the CF are drawn. Atmospheric temperature uncertainties have the largest impact on N2O retrievals in both B1 and B2 with CF a factor of 4 of more larger than for the other parameters in the mid-upper troposphere. As stated above, this means that the T and the N2O retrievals are not independent. Therefore high (low) N2O could be caused by high (low) T or the other way around but the impact cannot be determined because of the joint retrieval.

$\rightarrow$ Both B1 and B2 are sensitive to the temperature which the prior knowledge is from the operational level 2 products. Thus, we can either fix the temperature profile to these apriori profiles or co-retrieved the temperature profiles simultaneously with $N_2O$. This later strategy that we used in our study allows a global adjustment of the calculated radiances together with all the parameters that the forward model is sensitive to. However, the two retrieval strategies allow to assess the contamination on the target species. When the interfering parameter is fixed, the contamination can be quantified via the model parameter error (cf. Roders, 2000 page 48) and when it is part of the state vector it can be quantified via the extra-diagonal elements of the global averaging kernel matrix (cf. Rodgers and Connor 2003; Rodgers, 2000 page 70) as it is explained in the response #11 to the referee #1. The CF of T is larger than the CF of $H_2O$. However, since the variabilities of water vapor and therefore of its CF are larger than for the other parameters, the CF of $H_2O$ results in a higher impact on the $N_2O$ spatial and temporal variability especially over tropics. The figures 11, 12 and 14 clearly show this impact over tropics. Thus, this difference of behaviour between $N_2O\_B1$ and $N_2O\_B2$ especially over tropics shows that the parameter which impact the $N_2O$ variability is not T which CF is similar in B1 and B2.

iii-Emissivity retrieval: The authors state that in RTTOV the ocean emissivity is parameterized and that land emissivity is prescribed by an atlas. They call these emissivities a priori emissivities and retrieve surface emissivity in their procedure. Are the emissivities spectrally varying in RTTOV? How are the emissivity jacobians computed in RTTOV? Are they the same over sea and over land? It would be interesting to see results from emissivity retrievals and the differences over sea and land and over different types of land. Surface temperature and surface emissivity are parameters with signatures hard to discriminate in a small spectral window such as B1 or B2 as they basically give the background slope. The retrieval of both parameters is probably redundant. The authors should give information about how much the spectral chisquare has been improved when surface emissivity is retrieved and about the improvement it provides on the validation dataset. In case of no or too small improvements, the retrieval procedure has to be reconsidered without emissivity retrieval.

$\rightarrow$ In RTTOV, surface emissivity over land and ocean are spectrally varying. The computation of the emissivity Jacobian is similar as for the other parameters. It is computed by analysing the perturbation of the forward model for a given perturbation of the parameter assuming a linear relationship between these two perturbations ($\mathbf{\delta y = H(x_0)\delta x}$) . It is true that the spectral information of the surface temperature and the emissivity are difficult to discriminate. In our retrieval, the surface temperature gives the most reliable physical information about the surface since the apriori profile and errors are taken from the IASI level 2. Thus, in our retrieval strategy, the emissivity behaves as a mathematical adjustment parameter since we took a

relatively high (~10%) apriori std errors.  In addition, according to the figure 2, the spectral impact of the surface parameters (represented by surface temperature) is limited (but not negligible).  The emissivity is therefore simply used as a by-product in our retrieval strategy and is not dedicated to being scientifically analysed as a retrieval product.

In addition, the value of the cost function xhi2 (which is neither a necessary and nor a sufficient condition for the quality of a retrieval) does not vary significantly with and without the retrieval of the emissivity. The spectral variation of the surface emissivity here contributes to stabilise the inversion scheme by mathematically "absorbing" some spectroscopic transition which are not accurately taken into account in our fast-radiative transfer model.  This induced more realistic and stable vertical profiles. But this kind of methodology is very empirical and is hard to proved analytically.

iv-Validation: Equation 10 is applied to the HIPPO profiles to take the IASI vertical resolution and the impact of the a priori profile into account. Nevertheless, in order to apply this equation, the validation profiles have to cover the whole atmosphere. How and with what data are the tropospheric HIPPO profiles completed above the aircraft profiles top? How is the tropopause altitude taken into account? Concerning the comparison between the empirical and the theoretical errors there is a conceptual error. The authors compare the standard deviations of the differences between smoothed validation profiles and retrieved profiles (Emp) to the theoretical error (sum of smoothing and measurement errors Theoret) (Fig. 4). But as the validation profiles are smoothed by equation 10, the smoothing error is already taken into account and Emp has to be compared to RetNoise. As RetNoise is larger than Smooth this would not make a big difference. The other way is to compute the differences between the retrieved profiles and the raw validation profiles and to compare Emp with Theoret. Furthermore, the authors have shown that T uncertainty is largely impacting N2O retrievals (see CF) but as they retrieve jointly both parameters they cannot compute the resulting error. If the T profile was kept constant as suggested above, the errors caused T uncertainty could be evaluated (see Rodgers 2000). The errors caused by the other parameters should also be taken into account to compute the Theroret error but the same problem arises. The authors compute the Se matrix to provide the smoothing error instead of using Sa. Nevertheless Sa should represent the actual N2O global variability as accurately as possible and is the matrix that should be used to compute the smoothing error in equation 7 (Rodgers 2000). Se computed from the HIPPO data is representative of oceanic N2O profiles for given periods and may underestimate the variability. If the authors think it is a better representation of N2O global variability they have to justify this choice and may use it also for the retrievals. Furthermore a graphic representation of Se (diagonal values and covariance/correlation) should be given and compared to Sa.

→ The HIPPO profile has been extended using the chemistry transport model LMDz-Or-INCA. This has been clarified in the revised manuscript:

```
Using a similar method as explained in Kangah et al. (2017), we used
for these comparisons the measurements from the Harvard/Aerodyne
Quantum Cascade Laser Spectrometer (QCLS), one of the airborne
instruments of HIPPO, and the retrieved profiles selected within a
collocation temporal and spatial window of ±200 km and ±12h,
```

respectively. We extended the HIPPO profiles using monthly averaged profiles from the chemistry transport model LMDz-Or-INCA. To minimize the impact of this extension and since we are interested in the upper tropospheric $N_2O$ (cf. paragraph 6.2), we only took HIPPO profiles with a ceiling pressure less than 250 hPa and a bottom pressure greater than 400 hPa.

The tropopause altitude is then considered via the model.

There is no conceptual error concerning the comparison between the empirical and the theoretical errors. The retrievals are compared with the raw HIPPO profiles to estimate the empirical errors. So, the empirical error can be compared with the sum of the smoothing error and the retrieval noise. This has been clarified in the revised manuscript:

The theoretical covariance matrix of the total errors is then compared with an empirical total errors covariance matrix calculated using the raw (without applying the retrievals averaging kernels) HIPPO measurements and the retrievals along the HIPPO campaigns flight paths (namely the covariance matrix of the difference between HIPPO profiles and IASI retrieved profiles).

The theoretical basis of the CF has been explained in II-i and II-ii.

Concerning the use of Se instead of Sa for the smoothing errors, Rodgers (2000, page 49) explained the necessity of using a matrix which is more accurate than Sa to compute the smoothing errors. Since, the smoothing errors should also represent the loss of fine structures, a statistic of these fine structures must be used. Thus, Sa which is a reasonable constraint for the retrieval can be not enough accurately build to describe the statistics of these fine structures. That why we used the HIPPO in-situ profiles to build Se and estimate accurately the smoothing errors of the retrievals. HIPPO is mostly over the ocean but have a significant number of profiles over land. In addition, HIPPO $N_2O$ database gives the whole $N_2O$ tropospheric profiles both latitudinal variations (one of the dominant variation mode) and seasonal variations through the 5 HIPPO campaigns. Of course, Se is not perfect and maybe slightly underestimates the variations of $N_2O$ over land surface.

Se and Sa are therefore different matrices playing different roles in the retrieval process and characterisation. We can't use Se as apriori error covariance matrix because that will induce a dependency between the retrieval results and the validation data.

Instead of R we should have r2 which shows the percentage variation in the retrieved profile that is explained by the variations of the validation profile. Therefore R> 0.707 is needed to have more than 50% of the retrieved variability coming from the real variability. It is also important to have a comparison of the variability of the validation data and of the retrieved data. All this information (standard deviation of the differences, r2, variability) should be given synthetically with a Taylor diagram.

→ We agree with the reviewers concerning the fact that R2 should also be given, since it shows how much the linear regression with the reference datasest explains the distribution of our retrieval. So, we also mentioned this parameter in the revised version:

… N2O_B1 and HIPPO measurements are moderately correlated (the Pearson linear correlation coefficient R=0.42) with a low bias and standard

deviation (called hereafter std) error of -1.6 ppbv (~0.5%) and 3.5 ppbv (~1.0%), respectively. Thus, the linear regression using HIPPO measurements explains 18% ($R^2$=0.18) of the variations of $N_2O\_B1$.

…

The consistency between $N_2O\_B1$ and HIPPO increases at mid-latitudes (e.g. $R^2$=0.4 for northern hemisphere mid-latitudes). We can also notice that there is a very low mean bias (-0.1 ppbv) in the northern hemisphere high-latitude regions.

…

$N2O\_B2$ is moderately correlated with HIPPO measurements (R=0.6) with a std error of 3.2 ppbv and a very low mean bias of 0.3 ppbv. Thus, 36 % of $N_2O\_B2$ variations are explained by the linear regression with HIPPO measurements.

…

In tropical regions, the correlation coefficient between $N2O\_B2$ and HIPPO measurements becomes very high (0.71 and 0.92 in the northern and southern hemispheres, respectively) compared to the other regions. Therefore, in tropical regions more than 50% of $N_2O\_B2$ variations are explained by the linear regression with HIPPO measurements.

The figures 9 and 10 show scatter plots of our retrievals and HIPPO in-situ $N_2O$ measurements and synthetically give informations about the systematics errors (bias), the random errors (std error) and the accuracy of the variations (R and regression slope). We also give the variation explained by the linear regression with the reference datasets ($R^2$). This kind of plots are largely used in retrievals validation studies and therefore allow a quick comparison between different retrieval datasets. We think that there is no need to use a Taylor diagram here.

**III-Results:**

i-Validation: The retrieval results are not fully convincing. When the whole HIPPO dataset is used, meaning the strongest statistics (N about 100), r2=0.18 for B1 and 0.36 for B2 implying only 18 and 36% of the retrieved variability explained by the actual variability. Even if based on a limited HIPPO dataset, Garcia et al. (2017) achieve a better correlation (r2 = 0.58) whith a similar type of comparison as presented here. As they deal with a very close type of comparison, the results of Garcia et al. (2017) even in a paper under review should be discussed here. In most latitudinal bands (weaker statistics with N < 30) r2 is lower than 0.5 especially in the B1 case with a maximum of 0.4 in the northern mid-latitudes. In the B2 case r2 is the highest (0.85) for the tropical southern latitudes. But in that case it is based on 12 points only which makes the statistics really poor and the high R is due to the fact that the points are separated in to clusters. Furthermore, in the best r2 cases (tropical southern and northern latitudes for B2) the slopes of the linear interpolation are much larger than unity (2.5 and 3.3) indicating a largely too strong variability of the retrieved vmr's compared to the validation vmr's. For northerm mid-latitudes r2 = 0.4 for B1 and 0.29 for B2 which are rather low values. Finally, the authors state that in summary N2O_B1 and B2 are of sufficient quality to analyse N2O variations in the mid and high latitude regions. This conclusion is not really supported by the validation results as discussed above. Especially for high northern latitudes with r2= 0.1 for both B1 and B2, only 10% of the variability comes from the actual N2O variability. We would rather say that these data should not be used.

→ We do not agree with the referee concerning the quality of the retrieval. For N$_2$O_B1, at global scale R2 is about 0.18 (merging all latitudinal bands) and about 0.4 over mid-latitude regions. These results are consistent with those presented in Kangah et al., 2017 for GOSAT N$_2$O retrievals maritime pixels and in Garcia et al. (2018). In addition, from the comparison with HIPPO, N$_2$O_B1 present better accuracy (bias and σ) than the previous studies (Xiong et al.,2014; Garcia et al, 2018; Kangah et al., 2017). Results from Xiong et al., has been successfully used to derived global trends of N$_2$O and GOSAT Mediterranean N$_2$O have been linked to high N$_2$O emissions hotspots over Asia. This means that having a R2 < 0.5 does not mean that the data are not good enough to be used for scientific purpose. Therefore, our current N$_2$O_B1 product can at least be used for this kind of studies.

In addition, for N$_2$O_B2 the results of the comparison with HIPPO are better than all current validated N$_2$O profiles products in terms of R, R2, std errors and bias. Thus, N$_2$O_B2 can be used at finer time scale especially in tropical regions where R2 is better than 0.5 (R2=0.5 for northern hemisphere tropical regions over N=32 collocated pixels).

ii-Transport study: The variability of IASI N2O at 309 hPa shown on Fig. 13 is probably coming from a tropopause height difference. As shown by the AvK's, IASI vmr at 309 hPa is sensitive to a very large altitude range (600-120 hPa). Therefore it is equivalent to a N2O column or mean vmr over this range. When the tropopause changes from ~100 hPa in the tropics to ~250 hPa in the extratropics, the corresponding N2O columns mechanically change because the N2O vmr is lower in the stratosphere than in the troposphere. The authors attribute the N2O enhancement to upward transport from the Asian BL and horizontal transport within the anticyclone. This is also probably the case as shown by an extended literature based on satellite CO observations (Park et al., JGR, 2007...). Nevertheless, N2O is a well mixed gas and the quantification of such an effect is rather complicated. Surface in-situ data generally show a very limited seasonal variability of the N2O mixing ratio even in emission regions. Therefore the Asian BL is probably not N2O enriched as it is CO enriched. If the authors have evidence and data to document an important N2O enrichment during the monsoon in south Asia they should provide and discuss it. Another element that tends to strengthen the tropopause effect is that the IASI N2O high values are not limited to the anticyclone boundaries but to the whole tropical region. See in particular the high N2O band between 15 and 5◦N which is outside of the anticyclone (the southern boundary of the anticyclone is at about 15◦N). In order to have a better idea of the tropopause versus BL transport effects (I) the region of Fig. 13 should be extended both in latitude and longitude (ii) the boundaries of the anticyclone should be provided on Fig. 13 based for instance on PV values (see Ploeger et al., ACP, 2017) or on geopotential height values (e.g. Randel and Park, JGR, 2006). The Hysplit study is based on online simulations and simply shows that on the southern edge of the anticyclone, transport is westward which is expected. It does not prove that the air parcels are coming recently from the south Asian BL (the backtrajectories end up between 700 and 300 hPa and with a tenths of trajectories the statistics are very poor when Lagrangian studies are performed with millions of air parcels) nor that N2O enhancements over the whole tropical band could be due to such a transport process. The Hysplit part is therefore largely insufficient to draw conclusions and could be removed. The literature is rich enough about the subject of upward transport of BL air masses to the UTLS and trapping of pollution into the anticyclone. See for instance the Lagrangian modeling study of Bergman et al. (2013). References to this extended litterature are enough.

→ The emission and the transport of $N_2O$ from Asian BL to the Mediterranean Basin during the summer monsoon period has been largely addressed in Kangah et al., 2017. These high emissions during summer due, among others, to the high soil water content can be observed in current $N_2O$ emissions cadastre (e.g. cf. Kangah al., 2017, fig 9). The part 7 of the manuscript aimed to show that the spatial and temporal variation of $N_2O\_B2$ is consistent with this proved long-ranged transport structure despite the smoothing effect due to the shape of the averaging kernels (smoothing errors) and the retrieval noise.  Thus, over the period 21-23 July for instance, the figure 15 show a relatively homogenic tropopause level where the figure 14 show spatial variations of $N_2O$ over Asia with hotspots over eastern China and the Indian-Tibetan Plateau regions which were expected from the literature.  This cannot be explained simply by the tropopause effects.  The high $N_2O$ emissions are observed in most south Asian regions with hotspots over India and the north-eastern China. Thus, high $N_2O$ vmr are also expected in upper troposphere in all these regions in addition to the accumulation effect due to the monsoon anticyclone.

We agree with the reviewer with the fact that the hysplit part is not enough to draw conclusions. We used this part as an additional building block in our demonstration not as a self-sufficient assessment. This part is interesting, as it shows that the westward transport from Asia to the Mediterranean is consistent with daily $N_2O$ transport fluxes as represented by $N_2O\_B2$.

**IV-Minor comments:**

p2 l20-29: To my knowledge, the first paper to deal with tropospheric N2O retrievals from a satellite instrument is Chedin et al. (GRL, 2002). It shows very interesting results concerning the N2O evolution based on the TOVS instrument. This ref should be cited in the paper.

→ We have inserted a sentence relative to these observations.

```
Chédin et al. (2002) show the annual and seasonal variations
of N2O concentrations retrieved from the Television and
InfraRed Operational Satellite-Next generation (TIROS-N)
Operational Vertical Sounder (TOVS) instrument.
```

```
Chédin, A., Hollingsworth, A., Scott, N. A., Serrar, S.,
    Crevoisier, C., and Armante, R., Annual and seasonal
    variations of atmospheric CO2, N2O and CO concentrations
    retrieved from NOAA/TOVS satellite observations,
    Geophysical Research Letters, 29(8), 2002.
```

p3 l16: Turquety et al. (2004) does not concern IASI O3 retrievals. There are a number of recent refs concerning IASI O3 retrievals.

→ Done. We added the following reference in the revised manuscript:

```
Dufour, G., Eremenko, M., Griesfeller, A., Barret, B.,
LeFlochmoën, E., Clerbaux, C., Hadji-Lazaro, J., Coheur, P.-
F., and Hurtmans, D.: Validation of three different scientific
ozone products retrieved from IASI spectra using ozonesondes,
Atmos. Meas. Tech., 5, 611-630, https://doi.org/10.5194/amt-
5-611-2012, 2012
```

P4 l17-18: the authors should give a recent reference to justify their choice of NEDT.

→ Done.
We used for our retrievals NEDT from Clerbaux et al., 2009.

P5 l16: the authors should give a ref or a detailed explanation that justify the shape of their a priori covariance matrix. We also need information about the shape of the a priori matrices for the other retrieved profiles (are they diagonal?).

→ The apriori covariance matrix is derived and adapted from Rodgers, (2000, eq. 2.83). Concerning the shape of the whole matrix the manuscript has been modified as follows:

The a priori error covariance matrix $S_a$ is built for all chemical species and by considering parameters independent to each other as follows (cf. Rodgers, 2000):

$$S_{aij} = \sigma_a^2 \times \exp(-\left|\ln(P_i) - \ln(P_j)\right|) \qquad (3)$$

where $\sigma_a^2$ is an a priori variance error fixed for each parameter of the state vector and $P_i$ the pressure level at the level $i$.

Diagonal matrices are used for temperature profile and surface emissivity.

P6 l2: the choice of 30% for the a priori variability for H2O because of HDO is rather empirical and poorly justified. What does sink parameter mean?

→ cf responses #7 and #11 to the referee #1.

P6 l4 and l6: sensitivity studies are mentioned but the reader knows nothing about what they are made of. Details about the methodology used and about the results of these sensitivity studies are needed.

→ We removed this unclear expression since we fixed the apriori std errors using the estimated current knowledge about the variations of each parameter. The sensitivity studies are done via the figure 2 to decide which parameter should be retrieved or not.

P6 l14: ref for the radiometric noise (see above).
→ Done
Figures: Fig 14: this figure is of poor quality and should be improved. The winds should be superimposed such as on Fig. 13 in order to make a more straightforward comparison.

→ Done